# Diagnosis of fusion genes using targeted RNA sequencing

Erin E. Heyer [1], Ira W. Deveson [1,2], Danson Wooi[1,2], Christina I. Selinger[3], Ruth J. Lyons[1], Vanessa M. Hayes[1,2,4,5,6], Sandra A. O'Toole[2,3,6,7,8], Mandy L. Ballinger[7], Devinder Gill[9], David M. Thomas [7], Tim R. Mercer[1,2,10] & James Blackburn [1,2]

Fusion genes are a major cause of cancer. Their rapid and accurate diagnosis can inform clinical action, but current molecular diagnostic assays are restricted in resolution and throughput. Here, we show that targeted RNA sequencing (RNAseq) can overcome these limitations. First, we establish that fusion gene detection with targeted RNAseq is both sensitive and quantitative by optimising laboratory and bioinformatic variables using spike-in standards and cell lines. Next, we analyse a clinical patient cohort and improve the overall fusion gene diagnostic rate from 63% with conventional approaches to 76% with targeted RNAseq while demonstrating high concordance for patient samples with previous diagnoses. Finally, we show that targeted RNAseq offers additional advantages by simultaneously measuring gene expression levels and profiling the immune-receptor repertoire. We anticipate that targeted RNAseq will improve clinical fusion gene detection, and its increasing use will provide a deeper understanding of fusion gene biology.

[1] Genomics and Epigenetics Division, Garvan Institute of Medical Research, Sydney 2010 NSW, Australia. [2] St. Vincent's Clinical School, UNSW Australia, Sydney 2031 NSW, Australia. [3] Tissue Pathology and Diagnostic Oncology, Royal Prince Alfred Hospital, Sydney 2050 NSW, Australia. [4] Faculty of Health Sciences, University of Limpopo, Turfloop Campus, Mankweng 0727, South Africa. [5] School of Health Systems and Public Health, University of Pretoria, Pretoria 0002, South Africa. [6] Central Clinical School, University of Sydney, Sydney 2006 NSW, Australia. [7] The Kinghorn Cancer Centre and Cancer Division, Garvan Institute of Medical Research, Sydney 2010 NSW, Australia. [8] Australian Clinical Labs, Sydney 2010 NSW, Australia. [9] Department of Haematology, Princess Alexandra Hospital, Brisbane 4102 QLD, Australia. [10] Altius Institute for Biomedical Sciences, Seattle 98121 WA, USA. These authors contributed equally: Tim R. Mercer, James Blackburn. Correspondence and requests for materials should be addressed to T.R.M. (email: t.mercer@garvan. org.au) or to J.B. (email: j.blackburn@garvan.org.au)

Chromosomal rearrangements that juxtapose two different genes together can form a fusion gene. Fusion genes play a causal role in tumorigenesis, accounting for ~20% of human cancer morbidity[1]. However, the prevalence of fusion genes varies widely across different cancers, and many fusion genes are specific to certain cancer sub-types[1–3]. Accordingly, the rapid and accurate identification of fusion genes can characterise and stratify cancer diagnoses.

Precise fusion gene diagnosis can also inform subsequent therapeutic treatment, with several drugs having been successfully developed to inhibit fusion genes, including imatinib mesylate for treating *BCR-ABL1* and crizotinib for treating *EML4-ALK* fusion genes[4,5]. Fusion gene diagnosis can also predict prognosis, patient survival and treatment response[1,6,7].

Fluorescence in situ hybridisation (FISH) and quantitative real-time polymerase chain reaction (RT-PCR) methods have been predominantly used for fusion gene diagnosis. Though highly sensitive, these methods typically only test for the presence of a single fusion gene, often resulting in a lengthy, iterative and costly path to diagnosis. Furthermore, these methods are unable to identify novel fusion gene partners or resolve complex structural rearrangements. As a result, false-negative results attributed to non-tested or novel fusion genes and isoforms are a leading cause of misdiagnosis of haematological cancers[8].

RNA sequencing (RNAseq) can address many of these limitations by providing genome-wide surveillance of fusion genes with nucleotide-level resolution of fusion junctions. However, due to the sheer size of the transcriptome, RNAseq suffers from poor sensitivity for detecting fusion genes that are lowly expressed or diluted by accompanying non-cancerous cells within a sample[9,10].

We recently developed a targeted RNAseq method that uses biotinylated oligonucleotide probes to enrich for RNA transcripts of interest[11,12]. This method enhances sequencing coverage by targeting and capturing hundreds of genes within a single assay, enabling the sensitive detection of rare or lowly expressed transcripts. Given these advantages, targeted RNAseq has been proposed as a fusion gene diagnostic in solid tumours and lung cancer[13,14] (Fig. 1a).

Here, we evaluate the diagnostic power of targeted RNAseq for fusion gene detection. In this analysis, we demonstrate its ability to identify different fusion genes in a variety of sample types and measure the influence of different laboratory and bioinformatic variables on performance. We show that in a cohort of clinical patient samples, targeted RNAseq increases the diagnostic rate from 63 to 76% compared to FISH and RT-PCR methods. Finally, we explore the supplementary use of targeted RNAseq to profile the immune-receptor repertoire within a sample, measure expression of marker genes and identify novel exons.

## Results

**Design of panel to capture fusion genes.** We first designed an expansive panel of capture probes targeting almost all known fusion genes in cancer as manually curated from literature and publically available databases[1,3,15–33]. However, since the overall sensitivity of targeted RNAseq is inversely proportional to the sum of captured gene expression, we split the design into two panels to maintain high sensitivity while targeting all annotated exons for all genes. We created one panel for haematological malignancies (including leukaemia, lymphoma and myeloma) that targeted 188 fusion-related genes and one panel for solid tumours (including prostate, lung, sarcoma, ovarian and bladder) that targeted 241 fusion-related genes, with 43 genes targeted by both panels (Supplementary Fig. 1a and Supplementary Data 1, 2). Given their involvement in a range of fusion events in blood cancers, we also included the T-cell receptor (*TCRA/D, TCRB* and

*TCRG*) and immunoglobulin (*IGH, IGL* and *IGK*) loci on the blood panel (Supplementary Fig. 1a, b). Notably, the capture of these genes also allowed the simultaneous profiling of immune-repertoire expression within each sample. Although these designs were more expansive than those typically used in a diagnostic context, they facilitated a comprehensive investigation of clinically relevant fusion genes.

We also considered whether targeted RNAseq could simultaneously profile additional genes with prognostic and analytical value. Therefore, we included probes for 2 additional core transcription factors (5 also fusion-involved), 5 cell-type markers and 10 splicing factors on the blood panel[34–40] (Supplementary Fig. 1a, b). Similarly, the solid panel covered 14 immune genes that infer potential avenues of treatment (Supplementary Fig. 1a, c; personal communication with Australasian Sarcoma Study Group).

Finally, we added probes for sequencing spike-in controls. Both panels included probes for the External RNA Controls Consortium (ERCC) RNA spike-in controls, with the solid panel additionally containing probes for RNA spike-in controls that represent fusion genes (fusion sequins;[41] Supplementary Fig. 1a-c).

**Evaluation of targeted sequencing enrichment.** We initially evaluated the performance of the two panels by comparing targeted RNAseq to conventional RNAseq using matched RNA extracted from the K562 and RDES cell lines. We employed a double-capture approach to increase the on-target capture rate, achieving a mean 93% of reads aligning to targeted regions (compared to 4% of matched RNASeq libraries; Table 1). We also compared the abundance of ERCC RNA spike-ins between targeted and conventional RNAseq to precisely quantify the enrichment rate achieved by the capture, finding that targeted RNAseq achieved a mean 59-fold enrichment for the blood panel and 33-fold enrichment for the solid panel whilst maintaining quantitative accuracy and reliable detection down to 3pM input (Fig. 1b, c, Supplementary Fig. 2a, b). Notably, we detected minimal read coverage for the non-targeted ERCCs, indicating a lack of off-target contamination in our libraries (Fig. 1b, Supplementary Fig. 2a).

We next investigated the fraction of genes represented on the panel that were reliably tested using targeted RNAseq. Within both cell lines, we measured over 70% of targeted genes with expression above 15 transcripts per kilobase million (TPM; Supplementary Fig. 2c), observing broad and uniform read coverage across the full length of these expressed genes (Fig. 1d, Supplementary Fig. 2d). Furthermore, we found that splice-junction reads encompassed 77.8% of annotated introns on the blood panel and 84.6% of annotated introns on the solid panel (Supplementary Fig. 2e). Collectively, these findings suggest that translocations interrupting the majority of genes represented on the two panels would be detected with targeted RNAseq.

**Evaluation of fusion gene detection.** Following the successful validation of the targeted RNAseq panels, we next assessed our ability to diagnose fusion genes, utilising six cell lines (K562, RDES, 143B, GOT3, KARPAS45 and MLS1765-92) that harbour known fusion genes (Fig. 2a, Table 1). As reliable fusion gene detection with short-read sequencing is computationally difficult and relies on the identification of paired-end reads that span or overlap the fusion junction (Fig. 2a), we assessed a wide range of bioinformatic tools for fusion gene identification (reviewed in refs. [42–44]). Ultimately, we implemented a fusion analysis pipeline using *STARfusion* and *FusionCatcher*[45,46] (Supplementary Fig. 3). Due to the presence of numerous false positive fusion events, we required fusion genes to be detected by both algorithms. Using

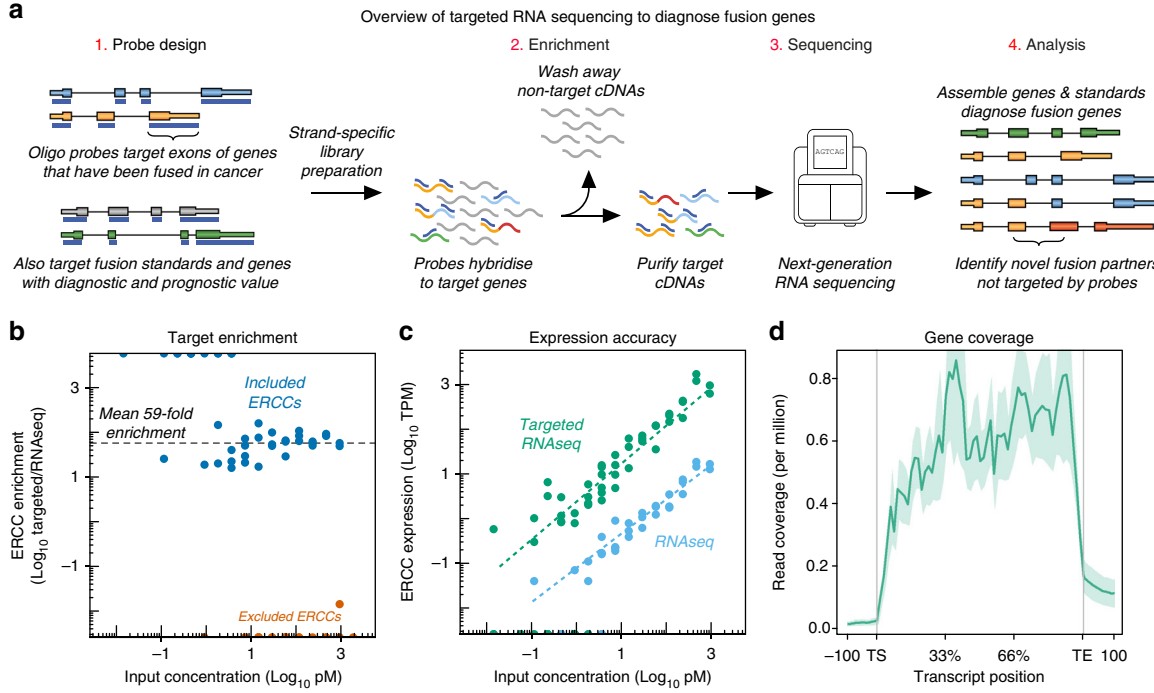

**Fig. 1** Overview of targeted RNAseq and panel validation. **a** Schematic of targeted RNAseq process. **b** Scatterplot of targeted RNAseq enrichment for ERCCs included on (blue) or excluded from (orange) the blood panel. **c** Abundance of captured ERCCs before and after targeted sequencing on blood panel. **d** Metagene plot of K562 targeted RNAseq read coverage across all genes on the blood panel. TS=Transcript Start site; TE=Transcript End site

### Table 1 Summary of cell line fusion genes and mapping statistics

| Panel | Cancer type | Sample | Detected fusion genes | Uniquely mapped reads (million) | On-target capture rate (%) |
|---|---|---|---|---|---|
| Blood | Bone marrow | K562 RNASeq | BCR-ABL1, NUP214-XKR3 | 46.0 | 3 |
| | | K562 | BCR-ABL1, NUP214-XKR3 | 10.7 | 98 |
| | | K562 1:10 | BCR-ABL1, NUP214-XKR3 | 49.7 | 72 |
| | | K562 1:100 | BCR-ABL1, NUP214-XKR3 | 4.9 | 91 |
| | | K562 1:1000 | BCR-ABL1, NUP214-XKR3[a] | 29.0 | 81 |
| | | K562 1:10,000 | BCR-ABL1[a] | 11.4 | 87 |
| | T-cell | KARPAS45 | KMT2A-FOXO4 | 16.9 | 97 |
| | WT | GM12878 | – | 10.4 | 98 |
| Solid | Sarcoma | 143B | EXOC2-MET, PAFAH1B2-FOXR1, ERG-LINC00240 | 27.5 | 92 |
| | | GOT3 | GPC6-WIF1, WNK1-ERC1, PPARD-IRF2BP2 | 27.1 | 93 |
| | | MLS1765-92 | FUS-DDIT3, CREB1-METTL21A | 20.1 | 93 |
| | | RDES RNAseq | EWSR1-FLI1 | 31.0 | 5 |
| | | RDES | EWSR1-FLI1, SMC04-EWSR1, FUS-DDIT3 | 30.4 | 88 |

[a]Indicates fusion gene identified by either STARfusion or FusionCatcher, but not both

this computational approach, we successfully detected known fusion genes in all cell lines (Table 1).

To measure the capture enrichment of fusion genes, we compared fusion junction read counts between targeted and conventional RNAseq. Although the *BCR-ABL1* fusion gene was easily detected in K562 RNASeq libraries (where the fusion gene is expressed from 8-24 DNA copies), the single-copy *EWSR1-FLI1* fusion gene was barely detected in the RDES cell line using standard RNASeq, illustrating the advantage of targeted RNASeq in fusion gene detection (Fig. 2b and Supplementary Fig. 4a, b).

Next, to assess the sensitivity of the capture panels for fusion gene detection, we prepared serial dilutions of K562 RNA from 1:10 to 1:10,000 against a GM12878 RNA background. Although we confidently detected the *BCR-ABL1* transcript in all samples through

to the 1:1000 dilution, it was only detectable with *STARfusion* in the 1:10,000 sample (Fig. 2c). Notably, this sensitivity is dependent on library depth, the number of genes captured and the fusion gene expression level, so may vary for different fusion genes.

Finally, to provide an absolute quantification of targeted RNAseq sensitivity in detecting fusion genes, we measured the detectable range of fusion sequins spiked into RNA extracted from the RDES cell line. We achieved 50% detection of fusion sequins at 2 pM input and 100% detection of all fusion sequins at their expected relative abundances between 8 pM and 31 nM input (Fig. 2d). Notably, this positive identification was independent of whether the panel targeted one or both fusion partners, demonstrating the ability of targeted RNAseq to capture and identify novel non-targeted fusion partners (Fig. 2d).

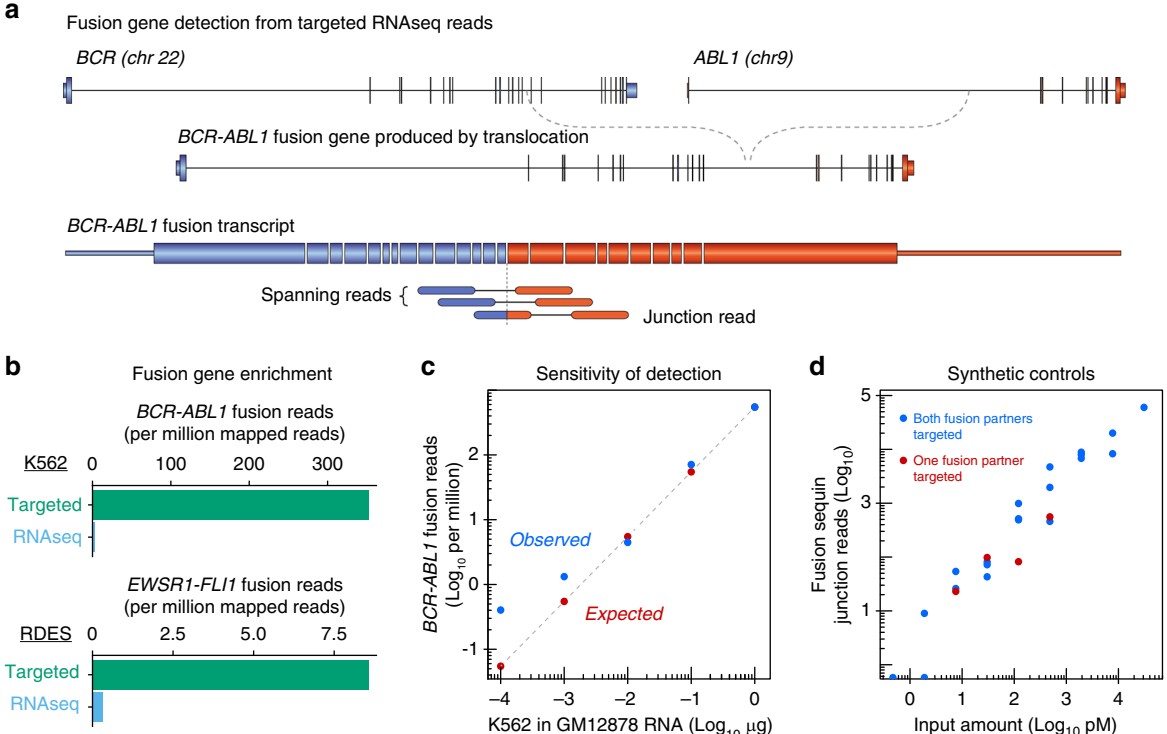

**Fig. 2** Validation of targeted RNAseq for fusion gene detection. **a** Diagram of *BCR-ABL1* fusion gene and transcript, depicting spanning and junction reads used to identify fusion genes. **b** Bar charts comparing abundance of fusion reads from targeted and canonical RNASeq libraries in K562 (top) and RDES (bottom) cell lines. **c** Scatterplot of observed (blue dots) and expected (red dots) *BCR-ABL1* read counts in K562 dilution series. **d** Scatterplot of fusion sequin junction reads versus input concentration

**Validation of fusion gene detection in clinical samples**. Following successful validation in cell lines, we next evaluated targeted RNAseq for fusion gene diagnosis in patient tumour samples. Initially, we assessed fusion gene detection in two lung cancer tumour biopsies previously diagnosed by FISH cytogenetics with break-apart probes (Fig. 3a, b). For each sample, library preparation and capture hybridisation were performed under clinical conditions within the St. Vincent's Hospital Research Precinct. In both cases, targeted RNAseq not only confirmed the previously identified *ROS1* and *ALK* rearrangements, but also ascertained both the fusion gene partners (*EZR* and *EML4*, respectively) and the precise fusion junction locations (Fig. 3d, e, and Supplementary Data 3).

We then expanded our analysis to test for the presence of fusion genes in a clinical cohort representing a broad range of cancer samples. In total, we profiled 72 samples encompassing 40 solid tumours using the solid panel and 32 haematological malignancies using the blood panel, as described above (Fig. 3d, Table 2). Patient-consented samples were collected by clinicians at St. Vincent's and Royal Prince Alfred Hospitals (Sydney), the Australian arm of the International Sarcoma Kindred Study (ISKS), the Kinghorn Cancer Centre Molecular Screening and Therapeutics (MoST) study and the Australasian Leukaemia and Lymphoma Group (ALLG) Discovery Centre.

Across the total cohort of 72 clinical patient samples, targeted RNAseq detected fusion genes in 55 samples (76%), a subset of which were validated by Sanger sequencing (Fig. 3d, Table 2, Supplementary Fig. 5f–k). In comparison, fusion genes were detected in only 39 out of 62 (63%) samples with prior molecular analyses (Fig. 3d, Table 2 and Supplementary Data 3). To specifically assess the overall concordance of these targeted RNAseq findings with previous diagnoses (ex. Figure 3a–c, Supplementary Fig. 5a–e), we compared the fusion genes

identified by both approaches. Targeted RNAseq correctly detected fusion genes in 33 out of 39 (85%) samples with previous fusion gene diagnoses, identifying both fusion gene partners in six samples where only one gene was previously identified (Fig. 3d and Supplementary Data 3). Of the six missed diagnoses, targeted RNAseq detected the inverse fusion gene in one sample and another was likely due to a promoter fusion event (see below). For the remaining 23 patient samples where previous molecular analyses reported no fusion genes, targeted RNAseq detected fusion genes in 12 samples (52%; Fig. 3d, Table 2 and Supplementary Data 3). Finally, targeted RNAseq identified fusion genes in 6 out of 10 (60%) patient samples where prior molecular testing reports were unavailable (Supplementary Data 3).

To measure the reproducibility of fusion gene diagnosis using targeted RNAseq in patient samples, we selected three samples – two with detected fusion genes, one without – and prepared targeted RNAseq libraries in triplicate to assess intra-run variability. These nine samples were also captured in triplicate and sequenced independently on three lanes to assess inter-run variability. We detected the expected fusion genes in all replicates of the two positive samples, whilst no fusion genes were detected in any of the negative sample replicates (Supplementary Data 4).

We next compared fusion junction read coverage between inter-run and intra-run replicates (Supplementary Fig. 6a, b). We observed low variability between inter-run and intra-run replicates with mean coefficient of variations of 0.073 and 0.071, respectively (Supplementary Data 4). In addition, we quantified the read coverage for every canonical gene on the capture panel and performed hierarchical clustering to illustrate the high reproducibility in gene expression measurements (Supplementary Fig. 6c).

We next assessed fusion gene diagnosis in these samples according to cancer type. Of the 20 prostate cancer samples

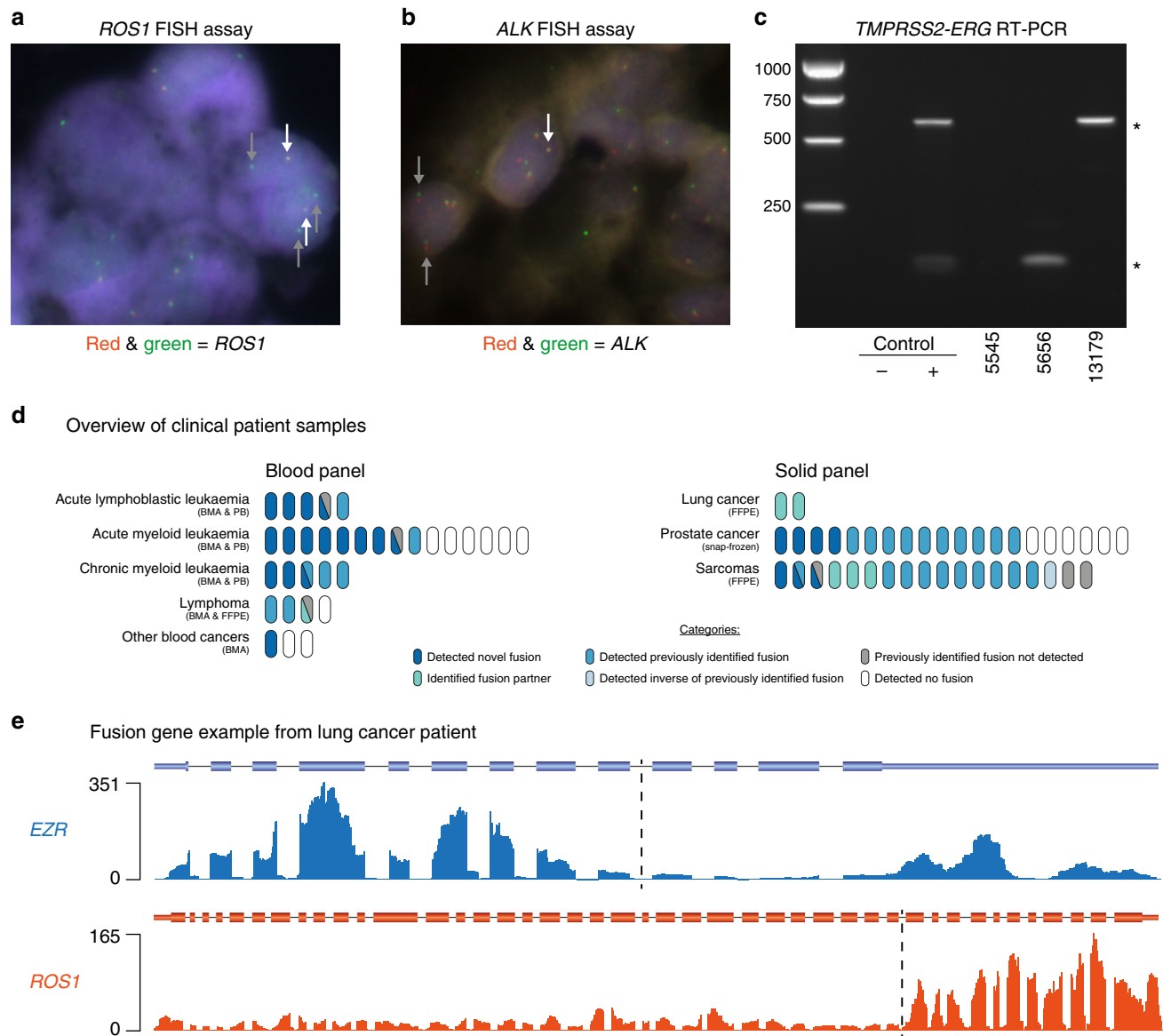

**Fig. 3** Fusion identification in clinical cohort samples. **a** FISH identification of *ROS1* rearrangement in lung cancer sample MO-16-000393. Positive signal is 1 fused set of red and green dots and ≥1 isolated green dots per cell. White arrows point to fused dots; grey arrows point to green dots. **b** FISH identification of *ALK* rearrangement in lung cancer sample SP-15-11000. Positive signal is 1 fused set of red and green dots, 1 isolated red and 1 isolated green dot per cell. White arrows point to fused dots; grey arrows point to isolated red and green dots. **c** RT-PCR analysis to diagnose *TMPRSS2-ERG* fusion genes in prostate samples. * indicates *TMPRSS2-ERG* bands. Source data are provided as a Source Data file. **d** Overview of fusion gene identification in all clinical cohort samples; each oval represents one patient. Other blood cancers includes chronic lymphocytic leukaemia, multiple myeloma and uncategorised blood cancer patients. BMA=bone marrow aspirate; PB=peripheral blood; FFPE=formalin-fixed paraffin-embedded. **e** Read coverage across *EZR* and *ROS1* genes in lung cancer patient sample MO-16-000393. Dotted line marks fusion junction of *EZR-ROS1* fusion gene

within the cohort, we confirmed all 10 (100%) samples previously diagnosed by RT-PCR and found fusion genes in an additional four samples (Fig. 3d, Supplementary Fig. 7a–c and Supplementary Data 3). The cohort also included 17 sarcoma patient samples with a prior molecular diagnosis, of which we confirmed seven (44%) samples with high-confidence fusion genes and six (38%) samples with fusion genes identified by a single fusion-finding algorithm, finding one (6%) sample where we identified the inverse of the fusion gene previously identified and one sample (6%) where we detected a novel fusion gene (Fig. 3d and Supplementary Data 3). In addition, we identified a novel fusion gene in one sarcoma sample lacking a previous molecular diagnosis (Fig. 3d and Supplementary Data 3).

Using the blood panel, we applied targeted RNAseq to analyse five acute lymphoblastic leukaemia (ALL) samples. This confirmed prior analyses in one out of two (50%) samples and detected fusion genes in two samples (100%) where prior testing identified no fusion genes and one sample (100%) with no prior testing information. In the ALL sample where previous RT-PCR detected an *AFF1-KMT2A* fusion gene, targeted RNAseq identified the *KMT2A-AFF1* fusion gene in addition to a previously unknown *AFF1-MYC* fusion gene (Fig. 3d, Supplementary Figs. 5j, 7d and Supplementary Data 3). As all three genes reside on separate chromosomes, these two fusion genes likely result from a complex genomic rearrangement. Of the 15 acute myeloid leukaemia (AML) samples analysed, we confirmed

**Table 2 Fusion genes found within the clinical cohort**

| Panel | Cancer type | Fusion genes detected with targeted RNAseq | FISH & RT-PCR | Targeted RNAseq |
|-------|-------------|---------------------------------------------|---------------|------------------|
| Blood | Acute lymphoblastic leukaemia | KMT2A-AFF1, AFF1-KMT2A, RUNX1-RUNX1T1, TCF3-PBX1, AFF1-MYC, TAF15-ZNF384, ZNF384-TAF15 | 2/4 | 5/5 |
| | Acute myeloid leukaemia | CBFB-MYH11, NSD1-NUP98, RUNX1-RUNX1T1, RUNX1T1-RUNX1, KMT2A-MLLT3, DEK-NUP214, NUP214-DEK, MN1-ETV6, ETV6-MN1, DDX3X-MLLT10, KMT2A-SEPT9, SEPT9-KMT2A, PML-RARA, RARA-PML | 2/9 | 9/15 |
| | Chronic myeloid leukaemia | BCR-ABL1, RUNX1-RUNX1T1 | 3/4 | 5/5 |
| | Lymphoma | MYC-IGH, IGH-BCL6 | 3/4 | 3/4 |
| | Other blood cancers | FGFR1-ZMYM2 | 0/1 | 1/3 |
| Solid | Lung | EZR-ROS1, EML4-ALK | 2/2 | 2/2 |
| | Prostate | TMPRSS2-ERG, ACSL3-ETV1, SP3-CTU2, SLC45A3-SKIL | 10/20 | 14/20 |
| | Sarcoma | SS18-SSX1, SS18-SSX2/2B, FUS-DDIT3, DDIT3-FUS, EWSR1-ERG, EWSR1-FLI1, PATZ1-EWSR1 | 17/18 | 16/18 |

Columns on the right indicate the number of patient samples with a positive fusion gene diagnosis from prior clinical assessment or targeted RNAseq; discrepancies in total sample number reflect the lack of available clinical data

previously reported fusion genes in 1 out of 2 (50%) samples and identified a novel gene in the other sample with a previously reported fusion gene. Additionally, targeted RNAseq identified fusion genes in 3 out of 7 (43%) samples where prior testing identified no fusion genes and 4 out of 6 (67%) samples with no information on prior molecular analyses. We confirmed previously detected fusion genes in all three (100%) chronic myeloid leukaemia (CML) samples and identified fusion genes in 1 CML sample where prior testing identified no fusion genes and one sample with no analysis history available. Similarly, we confirmed all three (100%) lymphoma samples with prior fusion gene identification. Finally, we detected a novel fusion gene in one uncategorised blood cancer sample.

Across the solid and blood panels, there were 23 patient samples where previous analysis identified no fusion genes. Of these, we reported fusion genes in 12 (52%) samples. In eight of these samples, the identities of the genes partners in the fusion gene were different from those previously analysed with FISH or RT-PCR. However, in the remaining four samples targeted RNAseq identified fusion genes that were previously tested for but not reported by either FISH or RT-PCR. This could be due to the additional sensitivity of targeted RNAseq or a discrepancy between the isoforms detected by targeted RNAseq and those analysed by FISH or RT-PCR; for example, in one instance (AML patient 36EW), unusual RT-PCR banding prevented the fusion gene from being reported (Supplementary Data 3). Both the issues of incorrect gene choice and varying isoform usage demonstrate the benefit of interrogating hundreds of genes at once in a manner independent of fusion junction location.

In total, 37 unique fusion genes were identified across our clinical cohort (Table 2). The 72 clinical samples in this cohort were prepared from a variety of sources, including both solid tissue (fresh-frozen and FFPE) and liquid samples (bone marrow and peripheral blood), with samples representing a range of RNA qualities. Despite this variability in sample type and quality, we observed only small differences in alignment performance. All double-capture samples reported ≥89% of reads mapping to capture panel regions (Supplementary Fig. 8a). The capture of targeted regions was slightly higher for liquid samples than tissue samples (median 99.3 v 94.7, $p = 5.8 \times 10^{-16}$, Wilcoxon rank sum test). However, there was no significant difference in capture efficiency between FFPE and fresh-frozen tissue, indicating that even challenging FFPE tissue can be effectively analysed using targeted RNAseq (median 94.5v 95.4, $p = 0.50$, Wilcoxon rank sum test; Supplementary Fig. 8b).

A unique advantage of targeted RNAseq is the ability to resolve alternative fusion gene isoforms that may inform clinical action.

For example, across the five CML patients, we identified two previously described BCR-ABL1 isoforms that were associated with disparate responses to imatinib treatment[47,48] (Fig. 4a). The presence of multiple fusion transcript isoforms was most notable in the prostate cancer samples, where 10 out of 11 (91%) TMPRSS2-ERG positive samples expressed two or more alternative isoforms (Supplementary Fig. 9a). In total, we identified 10 distinct TMPRSS2-ERG fusion isoforms, with the majority exhibiting complex 5′ end diversity from alternative TMPRSS2 transcription start sites (Fig. 4b). We also detected multiple fusion gene isoforms that resulted from different translocations upstream or downstream of ERG exon 3, though these alternative isoforms had no effect on expression level (Supplementary Fig. 9a, b).

Across the entire clinical patient cohort, 24 out of 54 (44%) patient samples harboured fusion genes whose diagnosis would inform subsequent clinical action (Supplementary Data 3). Six (25%) of the actionable fusion genes were not previously identified using alternative methods (Supplementary Data 3). While some fusion genes, such as SS18-SSX1 and MYC-IGH, constitute prognostic factors, other fusion genes, such as EML4-ALK and PML-RARA are directly targetable.

**Measuring gene and exon expression with targeted RNAseq**. In addition to identifying fusion genes, targeted RNAseq simultaneously measures the expression of all captured genes within each sample[11]. Initially, we quantified read coverage for each exon and found that abrupt changes in read coverage corresponded to fusion junction locations (Fig. 4c, d). This likely represents the difference in overall expression levels between the fusion gene and the non-fused, canonical alleles, though observed expression levels will depend on the sum of expression of the fusion gene, the inverse fusion gene (in the case of balanced rearrangements), and any non-rearranged alleles. For the majority of patient samples, high fusion gene expression contrasted with little or no expression from the non-rearranged alleles, suggesting the existence of additional factors that lead to enhanced expression. For example, the EZR-ROS1 fusion gene was highly expressed compared to the corresponding, non-fused EZR and ROS1 genes (Fig. 4c). However, in a minority of cases, the endogenous expression of the 5′ fusion gene drives fusion gene expression. For example, the ACSL3-ETV1 fusion gene exhibited similar expression to the corresponding ACSL3 gene, which likely results from the translocation of the ACSL3 promoter and its regulatory activity (Fig. 4d).

Notably, for one sarcoma sample, targeted RNAseq was unable to identify a fusion gene, despite previous FISH analysis reporting

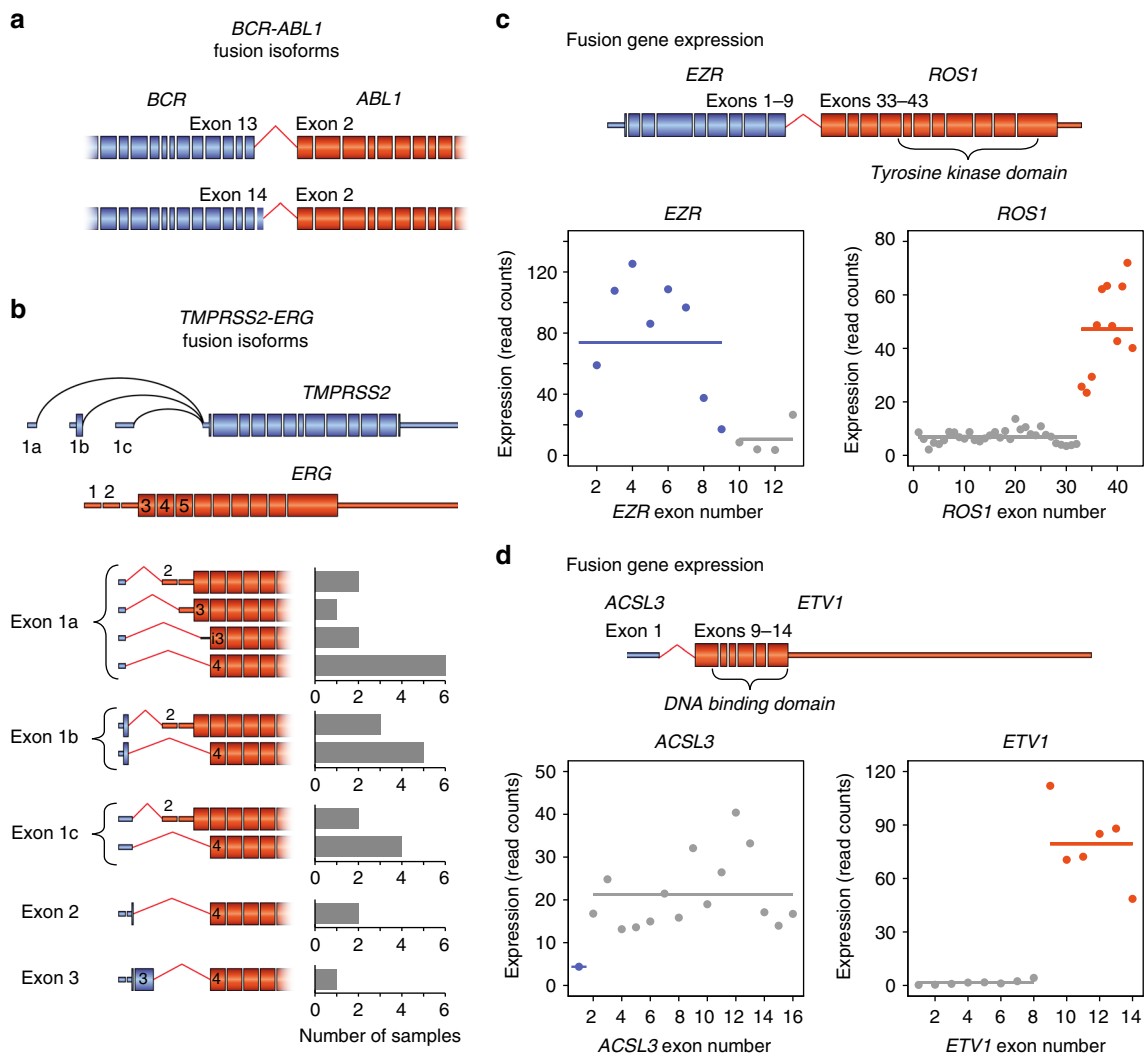

**Fig. 4** Fusion junction diversity and gene expression. **a** Schematic of *BCR-ABL1* fusion isoforms +/−*BCR* exon 14. **b** *TMPRSS2* and *ERG* gene structures and *TMPRSS2-ERG* fusion isoform prevalence. Bar charts on the right indicate the number of samples expressing each isoform. For simplicity, junctions beyond exon 1 are depicted utilising exon 1a. Black line represents retained intronic sequence. **c**, **d** Schematic of *EZR-ROS1* and *ACLS3-ETV1* fusions and quantification of read count expression across the endogenous genes in lung cancer sample MO-16-000393 and prostate cancer sample 12543, respectively. Horizontal lines indicate mean expression levels; coloured dots represent expression from the fused alleles plus nonrearranged alleles, while the grey dots represent expression of the canonical, nonrearranged alleles

a chromosomal rearrangement involving *ROS1* (Supplementary Data 3). Subsequent analysis of this sample showed *ROS1* expression to be 50-fold higher than the median of all sarcoma samples, supporting the existence of a promoter fusion that deregulated *ROS1* expression (Supplementary Fig. 10a, b). This suggests that whilst targeted RNAseq is unable to directly detect chromosomal rearrangements that fuse a promoter upstream of a different gene, it may still detect the resulting change in gene expression.

Finally, we expanded the gene expression analysis to the targeted genes that can yield cell marker or prognostic information. Whilst expression of these genes varied across samples, we nevertheless detected suggestive gene expression patterns. This was exemplified by high *GATA2* expression in some AML and CML patients, which is a known marker of poor prognosis in AML[49] (Supplementary Fig. 11, 12).

**Immune repertoire profiling**. As deregulated V(D)J recombination can create fusion genes involving IG/TCR receptor loci in a

range of blood cancers, our blood panel targeted the V, J and C exons at these loci (Fig. 5a). Accordingly, we identified three lymphoma patients within our patient cohort harbouring *IGH-MYC* or *IGH-BCL6* fusion genes. However, in addition to fusion genes, these probes also captured all RNA transcripts expressed from the immune receptor loci (Fig. 5a). Therefore, we next assessed our ability to resolve the immune repertoire profile within each sample.

We first captured RNA from B- (Daudi, Raji, Ramos) and T- (KARPAS45, Jurkat) cell lines with known V(D)J recombination events, as described above. We then used both *MiXCR* and *IMSEQ* to profile the clonotype population within each sample[50,51] (Supplementary Fig. 3). For each cell line, we detected 1–3 dominant clonotypes supported by the majority of immune reads, as expected for clonal cell lines (Fig. 5b and Supplementary Data 5). False-positive clonotypes were supported by only a small fraction of reads and predominantly derived from the same immune receptor loci.

Next, we extended this immune analysis to the 32 haematological patient samples (29 cancerous and 3 healthy) within the clinical cohort. In contrast to the cell lines, the majority of the

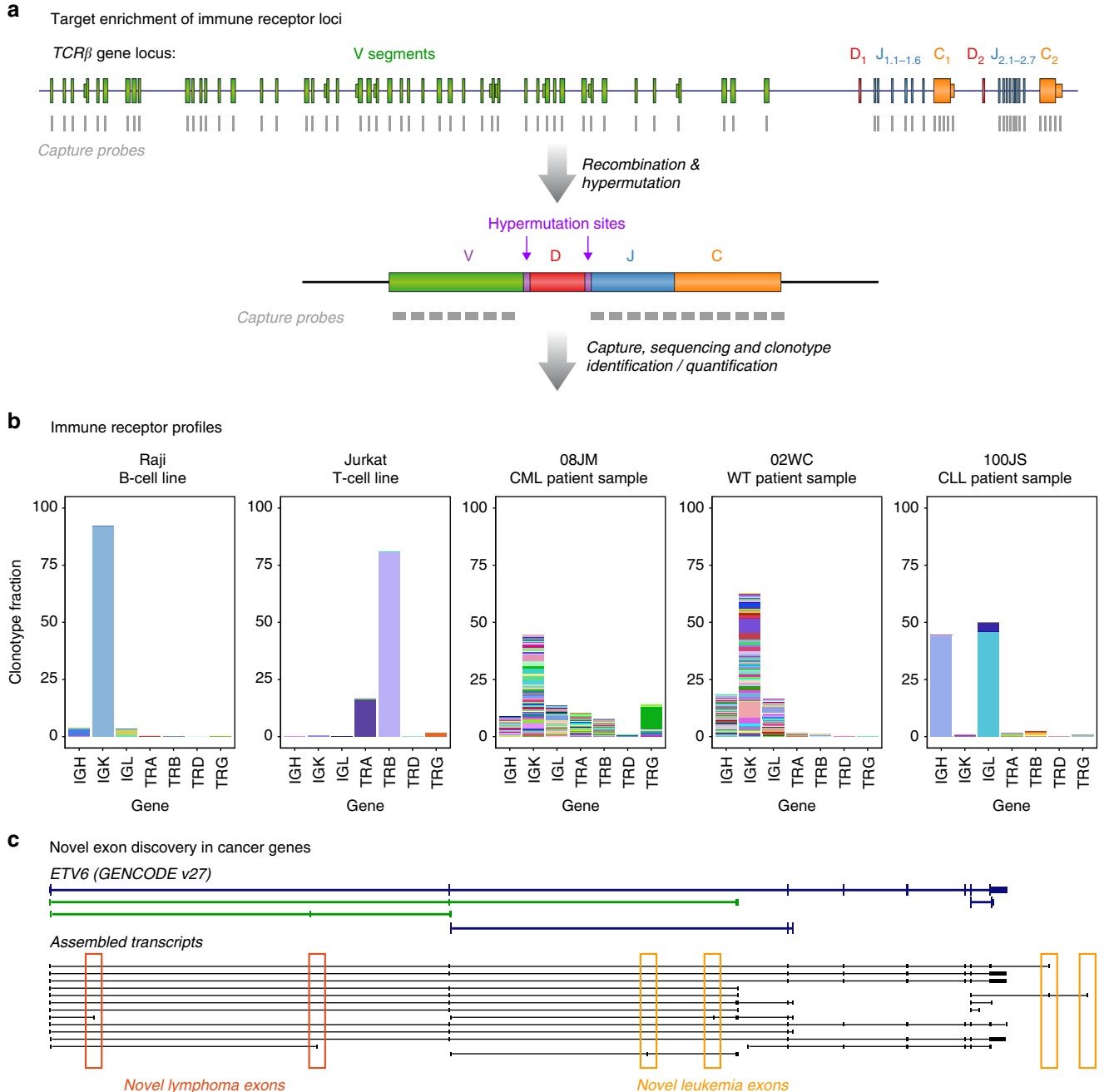

**Fig. 5** Novel findings in transcriptomic analysis. **a** Schematic of immune receptor capture probe design across the T cell receptor β (TCRβ) locus and a transcript expressed post-V(D)J rearrangement. **b** Immune receptor clonotypes in cell lines and clinical patient samples quantified using MiXCR. Each colour represents a single clonotype. **c** Novel *ETV6* exons shown underneath GENCODE v27 annotation. Red arrows indicate exons found in lymphoma samples, blue arrows indicate exons found in leukaemia samples

cancerous and healthy samples expressed hundreds of different immune receptor clonotypes, with each clone represented by a small number of reads (Fig. 5b and Supplementary Data 6). As expected for bone marrow aspirates, more IG clones were identified in each sample than TCR clones, reflecting the diversity of B-cells maturing in bone marrow (Fig. 5b). Notably, in 2 of the 29 cancerous samples, a set of T/BCR clones were ~10x and 100x more abundant than all other samples, possibly reflecting the presence of malignant T- and B-cell clonal populations (Fig. 5b and Supplementary Data 6).

**Novel transcriptomic features**. The enriched sequence coverage achieved by targeted RNAseq also enables the discovery of novel

exons and isoforms[11]. Given the clinical value of the genes targeted by our panels, newly discovered exons could become novel therapeutic targets. Therefore, we performed genome-guided transcript assembly to build an expansive annotation based on the clinical patient cohort. In total, we identified 528 novel exons within targeted genes, of which 256 were novel 5′ exons, 89 were novel internal exons and 183 were novel 3′ exons (ex. Fig. 5c).

To assess the validity of these novel exons, we investigated the flanking nucleotide composition for evidence of poly-pyrimidine tracts and 3′ splice site motifs. We found the flanking nucleotide profile of novel exons was similar to high-confidence exons annotated in GENCODE v27[52] and miTranscriptome[53] (Supplementary Fig. 13a). Additionally, novel exons exhibited a similar

size range to these previously annotated exons (Supplementary Fig. 13b). Although most (83%) novel exons encode alternative first or last exons, which may influence gene expression, we found that 70% of novel internal exons are predicted to modify the open reading frame (Supplementary Fig. 13c).

## Discussion

Chromosomal translocations that generate fusion genes are a major cause of cancer, and their accurate diagnosis is critical to effective treatment. However, previous methods such as FISH and RT-PCR rely on prior annotations, are low-throughput and limited in resolution. As a result, typically only the most common fusion genes are iteratively tested during diagnosis. Unfortunately, misdiagnosis in haematological malignancies can lead to delayed or unsuitable treatment[54].

In contrast to previous techniques, targeted RNAseq delivers high-resolution fusion gene detection whilst assessing hundreds of genes in a single test, identifying both known and novel fusion genes. This breadth can reduce time to diagnosis while improving diagnostic yield, exemplified by the novel fusion genes detected by targeted RNAseq that went undetected by prior molecular testing. The ability of targeted RNAseq to simultaneously identify multiple fusion genes in a single sample enables molecular stratification into cancer subtypes, while its use will also likely increase the catalogue of fusion genes – including rare fusion genes and novel gene partners – that are known to occur in cancer. Given these advantages, targeted RNAseq is increasingly being used for the diagnosis of fusion genes[14].

However, whilst the high-throughput nature of targeted RNAseq offers a broader path to diagnosis, it can also increase the false-positive rate at which fusion genes are detected. Indeed, this was a major challenge we faced, and our bioinformatic pipeline required supervision, manual curation and nuanced interpretation. This challenge may be offset by the development of high quality enterprise software or simultaneous analysis of matched-normal samples, which would indicate the prevalence of erroneous fusion gene calls and detect non-driver fusion events[55]. In addition, long-read sequencing can better resolve alternative fusion isoforms and would likely reduce spurious alignments that are a major source of erroneous fusion gene calls[56].

Targeted RNAseq also provides greater resolution of fusion gene loci. This includes the detection of chromosomal rearrangements that are complex and can only be ambiguously detected with other techniques. Furthermore, targeted RNAseq can resolve alternative fusion gene isoforms with distinct functional roles during disease development and treatment response. Indeed, we anticipate that isoform-level resolution of fusion genes using targeted RNAseq will ultimately provide more nuanced prognostic measures and better patient care[47,57].

Targeted RNAseq can also provide many supplementary benefits beyond fusion gene diagnosis. This includes the measurement of fusion gene expression and splicing that can predict treatment-resistance and variant detection to reveal the presence of treatment-resistant or cooperating mutations in signalling pathways[58]. The further measurement of gene expression signatures and markers can contribute additional prognostic information[59], whilst the ability to simultaneously resolve immunoglobulin and T-cell receptor clonotypes can detect the presence of B- and T-cell populations within a sample. We anticipate that this diversity of diagnostic features will be ultimately combined into a single unified targeted RNAseq test.

Although the spectrum of transcriptomic features that can be tested with targeted RNAseq will improve the breadth and value of diagnosis, this increased information will require careful interpretation to offset a greater risk for false-positive detection.

Nevertheless, such broad diagnostic measures will increase the likelihood of identifying treatable mutations for precision oncology. Accordingly, we anticipate that targeted RNAseq will be increasingly used - and eventually dominate current methods - for the diagnosis of fusion genes, leading to the improved diagnosis of cancer patients and further advancing our understanding of fusion gene biology.

## Methods

**Capture panel design**. Fusion gene content of the capture panels was based on extensive literature searches and through consultation with clinicians and pathologists; final gene lists are included in Supplementary Data 1 and 2. To ensure complete coverage of the T-cell receptor and immunoglobulin loci on the blood panel, we used previous PCR work as a reference[60] for mining all annotated IG and TR genes in both hg19 and hg38, including pseudogenes. Once the candidate target list was assembled and supplemented with ERCC and fusion sequin sequences, this was sent to Roche for proprietary SeqCap EZ design layout. For the canonical protein-coding genes, biotinylated DNA probes were tiled across all hg38-annotated exons from all isoforms with limited trimming of regions containing repetitive sequences or strong homology to other genes to minimise off-target results. Panels were assessed in silico against pre-existing RNAseq datasets prior to manufacture to ensure good coverage of all targets.

**Cell lines**. GM12878, K562 and KARPAS45 cell lines were sourced through the Coriell Institute, ATCC, and CellBank Australia, respectively. All were tested for mycoplasma and cultured according to standard growth protocols for each cell line. Cell lines were not independently verified. RNA was extracted from these samples following standard Trizol (Invitrogen) procedures. RDES, GOT3, 143B and MLS cell pellets were kindly provided by Maya Kansara for standard RNA extraction with Trizol. Total RNA from Daudi, Raji, Ramos and Jurkat cell lines was kindly provided by Joanne Reed.

**Patient samples**. Collection of patient samples was ethically approved: RPA X15-0103 and LNR/15/RPAH/143, ISKS Peter MacCallum Cancer Centre HREC Project Number 09/11, and MoST St Vincent's Hospital Sydney HREC/16/SVH/23. Additional patient samples were collected for this study under local Medical/Human Research Ethics Committee (MREC or HREC) approvals granted from the University of Limpopo's Medunsa Campus (MREC/H/28/2009) and the University of Pretoria's Faculty of Health Sciences (HREC#43/2010). Samples were shipped to the Garvan Institute of Medical Research under the Republic of South Africa Department of Health Export Permit, in accordance with the National Health Act 2003 (J1/2/4/2 #1/12). Analysis of the samples was performed in accordance with St Vincent's Hospital (SVH) HREC site-specific approval (#SVH15/227).

De-identified, patient-derived bone marrow aspirate and peripheral blood samples, frozen in Trizol, were sourced from the Australasian Leukaemia and Lymphoma Group (ALLG) Discovery Centre Melbourne. These samples were subject to ALLG Tissue Bank committee approval and accompanied by informed patient consent. The RNA was extracted according to Trizol manufacturers instructions, treated with TURBO DNA-*free* Kit (Thermo Fisher #AM1907) and purified using RNA Clean and Concentrator-25 columns (Zymo #R1017).

For all lung, prostate, SP-# sarcoma samples and all cell lines, Garvan Molecular Genetics (Sydney, Australia) extracted the RNA using the Qiagen QiaSymphony robot with associated reagents. For the remaining sarcoma samples, the FFPE samples were deparaffinised using Deparaffinization Solution (Qiagen, #939018), after which the RNA was extracted using the AllPrep DNA/RNA FFPE kit (Qiagen, #80234).

**Library construction**. Canonical RNASeq libraries were prepared using the Stranded mRNA-Seq Kit from Roche KAPA Biosystems (#07962193001) with inputs of 4 μg of RNA samples pooled with 1 μl of ERCC Mix 1 (Thermo Fisher #4456740). CaptureSeq libraries were prepared using the Stranded RNA-Seq Library Preparation Kit (#07227261001) with 100–1000 ng of RNA input plus 1 μl of ERCC Mix1 (except for the lymphoma samples and the Jurkat cell line, which were mixed with 1 μl of ERCC Mix2). Some solid samples contained additional 1 μl spike-ins of 1:50 dilution of fusion sequins[41]. Library construction followed manufacturers instructions using supplied reagents and Roche SeqCap adapters (#07141530001 and #07141548001) prior to 8–12 PCR amplification cycles, depending on RNA input. In some instances, homemade Y-adapters containing 1 out of 96 unique molecular identifier (UMI) barcodes were ligated to each end of dsDNA fragments following second-strand synthesis. These 8 nt UMIs were generated with the EDITTAG suite[61] using a Levenschtein editing distance of 4 and passed filters to remove homopolymers, 40% < GC-content < 60%, and sequences with complementarity to Roche adapters or indexing sequences.

**cDNA capture**. After library preparation with the Stranded RNA-Seq Library Preparation Kit (described above), samples were processed on the capture panels following the Roche-NimbleGen standard double-capture protocol (except for four

samples – 3x FFPE lymphoma and Jurkat, where a single-capture approach was used), as described in the SeqCap EZ Library support literature ("NimbleGen SeqCap EZ User's Guide [http://netdocs.roche.com/PPM/SeqCapEZLibrarySR_Guide_v3p0_Nov_2011.pdf]" and "Double Capture Technical Note [http://netdocs.roche.com/PPM/Double_Capture_Technical_Note_August_2012.pdf]". In brief, libraries, probes and Roche hybridisation reagents (SeqCap EZ Accessory Kit v2 #07 145 594 001; SeqCap EZ Developer Enrichment Kit #06 471 684 001; SeqCap EZ Hybridisation and Wash Kit #05 634 261 001; SeqCap HE-Oligo Kit A #06 777 287 001; SeqCap HE-Oligo Kit B #06 777 317 001) were incubated overnight at 47 °C. Libraries were washed and then re-hybridised for an additional overnight step to further enrich the subsequent capture libraries.

**Sequencing.** All libraries were sequenced on an Illumina HiSeq 2500 v4.0 platform at the Kinghorn Centre for Clinical Genomics (KCCG) in Sydney, Australia using a paired-end, standard depth 125 nt run.

**Panel validation.** Reads were barcode sorted by the sequencing facility to separate individual samples. When UMI-containing adaptors were used, paired-end FASTQ files were processed with Tally[62] to remove PCR duplicates, after which the UMIs were removed with cutadapt v1.14[63]. All reads were trimmed of Illumina adaptor sequences using cutadapt.

Sequencing reads were mapped to hg38 with STAR 2.4.2a_modified[64] using the default parameters with the following modifications: '--twopassMode Basic --outSAMstrandField intronMotif --outFilterMultimapNmax 100 --outFilterMismatchNmax 33 --seedSearchStartLmax 12 --alignSJoverhangMin 15 --outFilterMatchNminOverLread 0 --outFilterScoreMinOverLread 0.3 --outFilterType BySJout --outFilterIntronMotifs RemoveNoncanonicalUnannotated --chimSegmentMin 15 --chimJunctionOverhangMin 15 --alignMatesGapMax 200000 --alignIntronMax 200000'. All further panel validation analysis was limited to uniquely mapping reads, filtering for a mapping score of 255 using SAMtools[65].

On-target reads were identified using BEDTools[66] pairToBed to select the reads where at least one of each paired reads overlapped with the capture panel. Then, these on-target reads were normalised to the total number of uniquely mapping reads to calculate on-target capture rate.

TPM abundance and relative enrichments of each gene and spike-in were calculated using RSEM[67], while read counts per gene were calculating with htseq-count[68] version 0.6.0 using parameters '--stranded = reverse --type = exon --idattr = gene_id --mode = union'.

To calculate splice-junction reads covering annotated introns, we first isolated the mapped reads spanning introns by filtering for reads with a 'N' in the CIGAR string. These BAM entries were converted to BED format retaining the intronic region and then overlapped with existing intron annotations using BEDTools intersect with parameters '-s -F 1'.

**Fusion detection.** Trimmed and de-duplicated reads were used to identify fusion genes. FusionCatcher version 0.99.6a beta[46] was used with standard settings. Reads aligned with STAR (as above) were input to STARfusion[45]. As STARfusion and FusionCatcher often reported multiple fusion genes per sample, many of which were false positives, we added a number of filtering steps to increase our confidence in the fusion calls. First, we restricted the fusion candidate list to those where are least one of the fusion gene partners overlapped with the capture panel. Second, fusion gene calls were removed if they matched a manually curated blacklist (Supplementary Data 7) of fusion genes found in every sample (we noted that the identity of the false-positive fusion calls were predominantly software-specific and that these fusion genes were often specific to sample type). Third, we required each fusion gene to be supported by at least 2 reads, and the fusion junctions to be at least 10,000 nts apart if both genes were located on the same chromosome. Fourth, we filtered the STARfusion and FusionCatcher lists to select the fusion genes found by both programs, searching for overlapping fusion chromosomal coordinates. Finally, we manually curated these lists to separate high-confidence fusion genes (Supplementary Data 3) from false positive fusion genes (Supplementary Data 8), influenced by fusion genes with strong number of supporting reads and genes known to be active in the cancer subtype specific to each sample. For those samples where no overlapping fusion genes were identified, we manually searched through the output from both algorithms for known fusion genes, paying specific attention to fusion genes reported in the specific tumour type, to ensure that no fusion genes were overlooked.

**In-gene coverage change.** For each gene, the GTF entry for the main transcript isoform was extracted from the hg38 GTF file using grep and then converted to a BED file. The number of read 5′ ends falling within each exon were counted using BEDTools coverage and normalised to exon length to calculate expression.

**Transcriptome assembly and novel exon identification.** Following STAR mapping, as described above, only on-panel, uniquely mapping reads were input to Stringtie v1.3.3b[69] using parameters '--rf -f 0.05 -a 20 -j 3', guiding the assembly with a custom annotation file combining the latest annotations - GENCODE v27 GRCh38.p10[52] and miTranscriptome[53]. After transcript assembly for each patient

sample, the resulting transcriptomes were first combined with 'stringtie --merge' by cancer type and then merged across cancer types into a single representative cancer transcriptome. All further analysis was limited to multi-exon transcripts.

Exons were classified as novel if there was no genomic overlap with the GENCODE+miTranscriptome annotations, identified using BEDTools intersectBed with the 'intersectBed -v' option. Novel exons within targeted transcripts were identified using BEDTools intersectBed to select for any assembled transcript that overlapped with the annotated target gene.

**Immune receptor analysis.** After initial read trimming and removal of PCR duplicates, as described above, immune clonotypes were determined with IMSEQ v1.1.0[51] using standard parameters and MiXCR v2.1.3[50] using standard parameters, except for using '-OvParameters.geneFeatureToAlign = VRegion' during the initial alignment step.

**FISH.** FISH was performed on interphase nuclei on 3 μm formalin-fixed paraffin-embedded (FFPE) tissue sections using Vysis break-apart FISH probe kits (Abbott Molecular, Abbott Park, IL, USA). The FISH protocol was performed following the manufacturers' instructions, except that Invitrogen pretreatment solution (Life Technologies, Carlsbad, CA, USA) was used at 98–102 °C for 20 min. Image was cropped from larger image for publication with no alteration of signal levels.

**RT-PCR and Sanger sequencing.** *TMPRSS2-ERG* was detected by RT-PCR using a forward primer located in exon 1 of TMPRSS2 and a reverse primer located in exon 6 of ERG (TMPRSS2_RT-f: 5′-CAGGAGGCGGAGGCGGA-3′; TMPRSS2: ERG_RT-r: 5′-GGCGTTGTAGCTGGGGGTGAG-3′), analysed on an agarose gel and detected with GelRed (Biotium, #41033). Positive control is VCap cell line; negative control is PC3 cell line. An uncropped gel image is available in the Source Data file.

For fusion gene validation, cDNA was prepared from 1 μg total RNA using standard SuperScript II (Invitrogen # 18064014) reaction conditions. PCR from 1 μl of cDNA was performed with standard reaction conditions using 300 nM each primer and KAPA HiFi HotStart ReadyMix (KAPA Biosystems #KK2602). PCR bands were analysed on a 2% agarose gel stained with GelRed, isolated and extracted using the Zymoclean Gel DNA Recovery kit (Zymo Research #D4001). Sanger sequencing was performed with PCR amplification primers by Garvan Molecular Genetics at the Garvan Institute of Medical Research, Sydney, Australia.

**Graphics.** Metagene plots were created using the ngsplot package[70] with genome-mapping reads and parameters '-G hg38 -R genebody -F rnaseq -SS same -L 100'. Gene structure figures are based on screenshots from the UCSC Genome Browser[71]. Nucleotide frequency plots were created using "WebLogo 3 [http://weblogo.threeplusone.com/]", plotting probability on the y-axis. Dendrograms and heatmap were generated using pheatmap version 1.0.12[72]. All other plots were created in RStudio[73] using ggplot2[74] and cowplot[75] packages. All plots representing the number of fusion reads were prepared using spanning and junction read counts from STARfusion.

**Reporting summary.** Further information on experimental design is available in the Nature Research Reporting Summary linked to this article.

## Data availability

Sequencing data have been deposited in the NCBI Sequence Read Archive (SRA) with the BioProject code "PRJNA484669". Data for the figures presented are available in the Supplementary Data files and the Source Data file. All other data are available from the authors upon reasonable request.

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

## Acknowledgements
For their project input and for providing cell line RNA, we thank Maya Kansara, the Australasian Sarcoma Study Group (ASSG), Joanne Reed and Mandeep Singh. For providing malignancy samples, we acknowledge the Australasian Leukaemia and Lymphoma Group Discovery Centre, funded by The Leukaemia Foundation of Australia and the Australian National Health and Medical Research Council (NHMRC), and the International Sarcoma Kindred Study, supported by the Rainbows for Kate Foundation, the Johanna Sewell Research Foundation, the ASSG and NHMRC grants APP1125042 and APP1103685. Funding provided by NHMRC grants APP1108254 (T.R.M. and J.B.) and APP1114016 (T.R.M.), NHMRC Project grant APP1103685 (E.E.H.), NHMRC PRF APP1104364 (D.M.T.), Cancer Institute NSW CDF171109 (M.L.B.), Cancer Institute NSW Early Career Fellowship 2018/ECF013 (I.W.D.), Australian Postgraduate Award scholarship (D.W.), National Breast Cancer Foundation (S.A.O.), Sydney Breast Cancer Foundation (S.A.O.), and philanthropic donations from the Paramor Family (T.R.M.), from the Tag family foundation, the O'Sullivan Family, ICAP and Mr David Paradice (S.A.O.), and in memory of Domenico Marrocco (T.R.M.). The contents of the published material are solely the responsibility of the administering institution, a participating institution or individual authors and do not reflect the views of NHMRC.

## Author contributions
T.R.M. and J.B. conceived the project. V.M.H., S.A.O., M.L.B., D.G. and D.M.T. provided patient samples and clinical data. E.E.H., D.W. and J.B. performed RNA extractions, library preparation and targeted sequencing. C.I.S. and R.J.L. performed FISH and RT-PCR diagnostic experiments, respectively. E.E.H. performed Sanger sequencing validation experiments. E.E.H. and I.W.D. performed bioinformatic analysis. E.E.H., T.R.M. and J.B. wrote the manuscript with input from all authors.

## Additional information

**Competing interests:** T.R.M. was a recipient of a Roche Discovery Agreement (2014). The remaining authors declare no competing interests.

