## [Peer Review File · Nature Communications]

Reviewers' Comments:

Reviewer #1:

Remarks to the Author:

In their manuscript entitled "Diagnosis of fusion genes using targeted RNA sequencing" Heyer et al. describe the evaluation of a targeted RNA-sequencing assay designed to detect gene fusions in hematologic and solid tumors. The authors demonstrate that the targeted RNA-seq assay is able significantly increase the number of fusions that are detected. In addition, they demonstrate that the assay is able to provide gene expression and immune profiling data that might have clinical utility.

Major concerns:

- 1) A Table is needed that shows the canonical RNA-seq, targeted RNA-seq, and other test results (e.g. FISH or RT-PCR) for every sample/patient included in the study. This table should also include tumor type tested, the 5' and 3' genes involved in the fusion, the exons involved, and the genomic coordinates of the fusion partners. Without this it is hard to get a feel for the data.
- 2) The author's state that they have assessed the sensitivity and specificity of the assay but it is not clear what the gold standard was for sensitivity determination for example (sounds like it was FISH or RT-PCR for some or maybe all). That would be clearer with the table described above.
- 3) The authors stated there was a problem with false positive results. Were a large number of normal specimens run through the assay to help determine specificity? How often did you see false positive fusions in normal samples or tumor samples where you were not expecting to see a fusion. Are all fusions identified with the targeted RNA-seq assay reconfirmed with RT-PCR on a second sample.
- 4) It is stated that the assay detected an IGH-MYC gene fusion.
I am puzzled about an IGH-MYC fusion being detected by RNA-seq. The t(8;14) translocation juxtaposes the two genes but does not create a fusion transcript. See next to last paragraph of Discussion in article by Cleynen A et al, Expressed fusion gene landscape and its impact in multiple myeloma, Nature Communications, 8:1893, 2017 and Introduction of paper by Yan Y et al, Genes Chromosomes Cancer, 46:950, 2007. Also, I cannot find evidence of IGH or MYC gene fusions in the COSMIC database. What was the actual structure of this gene fusion (i.e. what exons were fused between the two genes).
- 5) From a clinical laboratory point of view a "systematic evaluation" (line 71) would include evaluation of not just accuracy but also reproducibility (e.g. intra-run and inter-run reproducibility, analytical sensitivity (i.e. LOD and LOQ) and assessment of impact of RNA quality on the results.
- 6) It is stated that only 2 supporting reads were required to call a case positive for a fusion. Did the authors explore different cutoffs to see if that would reduce the number of false positive calls.
- 7) How many fusions were generally observed per case, can you provide a range, average, median.
- 8) On line 211 there is a mention of a double capture approach but I could not find anything in the methods section or elsewhere in the paper that describes what is meant by this.
- 9) The paper describes a number of very interesting findings but seems a little too broad and unfocused. Might the authors for example describe the gene expression findings for the targeted RNA-seq assay in a second paper and just focus on gene fusion detection in this paper?
- 10) In the Discussion (line 317) it stated that nuanced approach is taken to the interpretation. Is that done at a post-analytical level when trying to ascertain the clinical significance of the identified fusion (which I can understand) or the analytical steps (which would be concerning).

Minor concerns:

- 1) In the Discussion (line 305) it is stated that "misdiagnosis is a leading cause of mortality in hematological malignancies" sounds a bit exaggerated to me.
- 2) Supp Fig 1: why were fusion sequins not included in the hematologic panel?
- 3) Supp Fig 2d: what do TS and TE stand for on the x-axis. Not sure if I totally understand this plot.

4)Supp Fig 7 figure legend should indicate that FFT stands for fresh frozen tissue

Reviewer #2:

Remarks to the Author:

In the article "Diagnosis of fusion genes using targeted RNA sequencing", the authors describe the application of targeted RNA-Seq to identify fusion genes in cancer samples. Accurate fusion gene detection is a highly relevant topic, both with implications for research and with translational value due to its diagnostic relevance. The authors propose that their targeted RNA-Seq design helps to overcome some of the limitations and has the potential to become a dominant method for detection of gene fusions in a clinical context.

The article is well written and has a clear structure. The authors demonstrate that targeted RNA-Seq is sensitive to identify most gene fusion events. The authors further show that their design can be used to obtain additional insights, for example into the expression levels of fusions genes, non-fusion genes on the panel, and it can profile the immune receptor loci.

The manuscript has a strong focus on the technology, whereas findings from the data are mostly confirmative, or proof of principle. While the authors acknowledge in the discussion that false positives is the major challenge, they do not show a systematic evaluation in the main text. Figure 2 shows examples that suggest the methods is sensitive, but there is no evaluation of specificity, considering false positives for the varying thresholds. The method to identify fusion genes requires manual curation, which prevents a systematic evaluation, and which provides a major barrier for any application outside of a research environment.

Major comments:

1) Systematic evaluation of sensitivity and specificity

As the technology is the center of this manuscript, it should be systematically evaluated. Increasing the number of genes compared to FISH will increase both sensitivity and false positives (more genes can be detected). Both needs to be evaluated. What is the false positive rate for single genes (e.g. a direct comparison with single gene methods)? What is the false positive rate when the full array is used? This will be important to understand whether this technology can be applied in a clinical context.

2) Implementation of a transparent and reproducible informatics workflow

The targeted RNA-Seq technology requires both sequencing and the analysis, and the evaluation should therefore be based on a complete, transparent, and reproducible workflow. In particular, manual curations or filtering should not be included in the evaluation. Manual curation is acceptable to avoid false positives when biological results are reported, but evaluation of a technology should be based on the bioinformatics workflow without manual interference/curation. The workflow should be completely reproducible from the manuscript.

3) The design could be explained in more detail.

A certain set of genes was selected, but which isoforms/exons/exon-junctions are used? Are known breakpoints included? Are all isoforms captured? If breakpoints are not enriched for, is there a loss of sensitivity to detect fusion genes compared to non-fusion genes?

Specific comments:

Figure 2b, what is the meaning of "On-panel"?

Figure 2c, a fair comparison would be fusion reads over non fusion reads for the genes of interest. The total enrichment is much higher due to targeting certain regions, not necessarily due to the enrichment.

Figure 3d: What is the meaning of each point? One patient? What if multiple colors are assigned to one dot?

Figure 3e: The numbers are not entirely clear from the legend and table. A systematic analysis of sensitivity and specificity would be good.

Figure 4c: There are 2 fusion gene combinations (as there are 2 promoters). There are also non fusion genes (full length EZR and ROS1 from cells which have intact versions). I don't understand the claim that the fusion gene shows higher expression than the non fusion gene based on these plots, I can only see that fusion gene expression seems to be dominated by the EZR expression level. How can I see the expression of non-fusion EZR here?

Figure 5c/Novel transcriptome features: What is the confidence to detect new exons? Are they supported by multiple reads, multiple samples? How can the authors exclude that these are artifacts? Does the read alignment favor splice site sequences over non-splice site sequences, and could this influence the observed sequence properties for novel exons?

"Diagnosis of fusion genes": To me this sounds misleading as it implies a clinical context. Are the authors referring to "diagnosis using fusion genes" or "detection of fusion genes"? From the abstract it seems like the authors imply both together?

We would like to thank the referees for the effort they have invested in reviewing and critiquing our study, as we consider the manuscript to be improved as a result of their efforts. Below, we addressed each specific comment of the original reviews (boxed) and underlined text that indicates changes we made to the manuscript.

Summary of revisions:

- 1) We expanded **Supplementary Table 3** to summarise our assessment of targeted RNAseq fusion gene detection relative to alternative clinical approaches. This includes genomic coordinates of the fused exons and the methods previously used to identify fusion genes for each sample.
- 2) Sanger sequencing validation for 5 samples where the fusion gene was not validated by previous molecular diagnostics is now included in **Supplementary Figure 5**.
- 3) We added **Supplementary Tables 6** and **7** to improve workflow reproducibility. **Supplementary Table 6** lists the blacklisted fusion genes that were identified across all samples and subsequently removed during bioinformatic filtering. **Supplementary Table 7** lists the false positive fusion genes identified in patient samples, their abundance, genomic fusion coordinates and the reason they were classified as false positives.
- 4) Updated numbers of samples with previously detected fusion genes, as additional information from the ALLG Blood Sample Biobank revealed previous molecular diagnostics for 19 samples. All relevant information and quantifications have been changed throughout the manuscript, figures and tables.
- 5) We have tailored the language used throughout the text to highlight that the work presented here is a detailed proof-of-concept and meant to evaluate the performance of targeted RNAseq for fusion gene identification from patient samples. Our work is not intended to constitute a controlled clinical trial.
- 6) We have revised the figure legends to improve clarity and fully define acronyms.

Reviewer #1:

In their manuscript entitled "Diagnosis of fusion genes using targeted RNA sequencing" Heyer et al. describe the evaluation of a targeted RNA-sequencing assay designed to detect gene fusions in hematologic and solid tumors. The authors demonstrate that the targeted RNA-seq assay is able to significantly increase the number of fusions that are detected. In addition, they demonstrate that the assay is able to provide gene expression and immune profiling data that might have clinical utility.

Thank you for your positive summary of our work.

Major concerns:

1) A Table is needed that shows the canonical RNA-seq, targeted RNA-seq, and other test results (e.g. FISH or RT-PCR) for every sample/patient included in the study. This table should also include tumor type tested, the 5' and 3' genes involved in the fusion, the exons involved, and the genomic coordinates of the fusion partners. Without this it is hard to get a feel for the data.

To provide a clear summary of our patient data and results, we have expanded **Supplementary Table 3** to include the specific genomic coordinates of the fusion genes identified with targeted RNAseq and both the gene identity and the type of detection method(s) previously used. We also included the transcript subtype(s) targeted by RT-PCR (where possible) and a count of the number of molecular diagnostic tests each sample underwent. This additional information supports our claim that a single targeted RNAseq assay can identify fusion genes in samples that often undergo multiple tests before a positive result is achieved. We have included genomic coordinates instead of exon number, as genomic coordinates remain constant, whilst the same exon can be assigned varying exon numbers across transcript isoforms dependent on alternative exon usage.

In total, we analysed 39 samples with previous molecular diagnoses for fusion genes; targeted RNAseq validated fusion genes in 33 (85%) samples. Targeted RNAseq detected fusion genes in 12 (52%) of 23 samples without a previous fusion gene diagnosis from FISH or RT-PCR analyses.

2) The author's state that they have assessed the sensitivity and specificity of the assay but it is not clear what the gold standard was for sensitivity determination for example (sounds like it was FISH or RT-PCR for some or maybe all). That would be clearer with the table described above.

We evaluated sensitivity in patient samples by comparison to both FISH and RT-PCR. As suggested by the reviewer, this has been clarified and summarized in **Supplementary Table 3**. In addition, we have also measured the quantitative sensitivity of targeted RNAseq to detect low abundance fusion genes based on a titration of K562 cells and by the detection of low abundance synthetic fusion genes (fusion sequins).

3) The authors stated there was a problem with false positive results. Were a large number of normal specimens run through the assay to help determine specificity? How often did you see false positive fusions in normal samples or tumor samples where you were not expecting to see a fusion. Are all fusions identified with the targeted RNA-seq assay reconfirmed with RT-PCR on a second sample?

Our study included three healthy samples for which we do not expect to observe fusion genes, with the results summarized in **Supplementary Table 3**. In addition, we used targeted RNAseq to test numerous cell lines and synthetic controls, for which we know the true positive fusion genes. Using these samples, we were able to evaluate the supporting read number and detection frequency for true positive fusion gene relative to false positive fusion genes.

We also suggest that fusion genes pervasively found in every sample likely represent false positive results. For example, we find that almost every sample harbors the *PPP1CB-SPDYA* fusion gene (shown below in Figure 4). Closer inspection of these two genes shows they share overlapping loci and do not represent a *bona-fide* fusion gene. In total, we identify 34 fusion genes that are pervasively found across samples and likely represent false positive fusion genes. These pervasive fusion genes are therefore listed on our blacklist (**Supplementary Table 6**).

Regarding fusion gene validation, fusion genes in 32 patients identified by targeted RNAseq were validated via agreement with prior molecular diagnostics (see updated **Supplementary Table 3**). Of the novel fusion genes, we validated 9 fusion genes by RT-PCR (Figure 1 below) and 6 with Sanger sequencing, which is now included in **Supplementary Figure 5**.

Figure 1. RT-PCR of novel fusion genes identified in patient samples.

4) It is stated that the assay detected an IGH-MYC gene fusion. I am puzzled about an IGH-MYC fusion being detected by RNA-seq. The t(8;14) translocation juxtaposes the two genes but does not create a fusion transcript. See next to last paragraph of Discussion in article by Cleynen A et al, Expressed fusion gene landscape and its impact in multiple myeloma, Nature Communications, 8:1893, 2017 and Introduction of paper by Yan Y et al, Genes Chromosomes Cancer, 46:950, 2007. Also, I cannot find evidence of IGH or MYC gene fusions in the COSMIC database. What was the actual structure of this gene fusion (i.e. what exons were fused between the two genes).

We apologize for the typo – we detected the *MYC-IGH* fusion gene (not *IGH-MYC* fusion gene). We have validated the *MYC-IGH* fusion gene by RT-PCR, as shown in the gel image above. This RT-PCR returned the correctly sized band, however, it is faint.

5) From a clinical laboratory point of view a “systematic evaluation” (line 71) would include evaluation of not just accuracy but also reproducibility (e.g. intra-run and inter-run reproducibility, analytical sensitivity (i.e. LOD and LOQ) and assessment of impact of RNA quality on the results.

Given that we have not performed additional evaluation of intra- and inter-run reproducibility, we have we have clarified the introduction to read "Here, we evaluated the diagnostic power of targeted RNAseq for fusion gene detection".

6) It is stated that only 2 supporting reads were required to call a case positive for a fusion. Did the authors explore different cutoffs to see if that would reduce the number of false positive calls.

We evaluated different cut-off thresholds for the number of supporting reads required to call a fusion gene. We found that whilst increasing the read cutoff reduced the number of false positive calls in some samples, it also reduced the number of true positive fusion genes detected in low quality FFPE samples. Therefore, we implemented a relatively low read cutoff, as we found that additional subsequent filtering steps omitted these false positives whilst not adversely impacting sensitivity.

To illustrate this, we plotted fusion junction reads normalized by library size for all fusion genes in the blood samples (Figure 2 below). This indicates that false positive fusion genes (shown in grey) cluster below 1 fusion junction read per million mapped reads (horizontal line), and there is an abundance difference between our reported fusion genes (in red) and the false positive fusion genes.

Figure 2. Scatterplot of normalized fusion junction reads for each blood sample.

The data plotted below are included in **Supplementary Table 7** to help our readers understand our filtering approach. This table includes a) the identity of these false positive fusion genes, b) their genomic coordinates and c) their abundance and d) the reason they were discarded.

7) How many fusions were generally observed per case, can you provide a range, average, median.

Of the 72 patient samples we analysed, targeted RNAseq reported 0 to 3 high confidence fusion genes in each sample, with a mean of 0.99 fusion genes per sample and a median of 1 fusion gene per sample (see **Supplementary Table 3** for details).

8) On line 211 there is a mention of a double capture approach but I could not find anything in the methods section or elsewhere in the paper that describes what is meant by this.

The double capture approach mentioned on line 211 is described in detail in the Roche Technical note cited on line 402 in the original manuscript (now line 462), and also described briefly on lines 406-407 in the original manuscript (now lines 466-467). In a double capture experiment, the RNAseq library is hybridized to biotinylated oligonucleotide probes in two sequential reactions (rather than a single reaction), thereby increasing the specificity of the capture.

9) The paper describes a number of very interesting findings but seems a little too broad and unfocused. Might the authors for example describe the gene expression findings for the targeted RNA-seq assay in a second paper and just focus on gene fusion detection in this paper?

One of the major benefits of targeted RNAseq beyond fusion gene detection is that the panel also generates a range of additional transcriptomic data that may have supportive clinical value, as was highlight by Reviewer #2. Whilst we agree that we have not undertaken a systematic analysis of the clinical value of this additional data, our concern is that the supportive nature of these analyses would not be apparent in a second paper, nor would the data be sufficiently strong for a stand-alone manuscript.

10) In the Discussion (line 317) it stated that nuanced approach is taken to the interpretation. Is that done at a post-analytical level when trying to ascertain the clinical significance of the identified fusion (which I can understand) or the analytical steps (which would be concerning).

This nuanced interpretation was applied to the analytical step of filtering the default output from *STARfusion* and *FusionCatcher* into the reported high confidence fusion genes. This is necessary to maximize the identification of true-positive fusion genes while minimizing the identification of recurrent and spurious false positive fusion genes. These false positive fusion genes result from a) the abundance and complexity of gene splicing, b) the difficulty of resolving this complexity with short-read alignments, and c) the complexity, diversity and uniqueness of the structural variants that generate fusion genes. Given that the inherent complexity of the human transcriptome results in false positive fusion detection, we do not think it will be possible to implement an unsupervised automated analytical pipeline in the near future (although it may be improved with long-read sequencing).

More broadly, we would like to highlight that the requirement for manual curation of results at analytical stages is required in almost all current diagnostic techniques. For example, currently FISH techniques rely firstly on interpretation by a trained cytogeneticist and secondly by a pathologist who incorporates these cytogenetic findings with additional information including

tumour cell morphology and phenotype, IHC data, etc. before making the final diagnosis.

Similarly, clinical diagnosis of genetic mutations using whole genome and/or target sequencing currently requires interpretation by an experienced bioinformatician, particularly for structural variants (which cause fusion genes). In a recent publication, it was noted "manual inspection of somatic variants identified by automated variant callers (i.e., manual review) is an important aspect of the sequencing analysis pipeline and is currently the standard for variant refinement. Manual review allows individuals to incorporate information not considered by automated variant callers." (Barnell, EK et al. 2018. Standard operating procedure for somatic variant refinement of sequencing data with paired tumor and normal samples. *Genetics in Medicine*).

Minor concerns:

1) In the Discussion (line 305) it is stated that "misdiagnosis is a leading cause of mortality in hematological malignancies" sounds a bit exaggerated to me.

We have modified the text and updated our references; this statement now reads "misdiagnosis in haematological malignancies can lead to delayed or inappropriate treatment".

2) Supp Fig 1: why were fusion sequins not included in the hematologic panel?

Probes for the fusion sequins were not included in the hematologic panel merely due to timing; they had not been developed when the blood panel was designed and ordered. We aim to include the fusion sequins in future iterations of the blood panel, but since these panels are manufactured in bulk it is not feasible to add in the fusion sequins post-production.

3) Supp Fig 2d: what do TS and TE stand for on the x-axis. Not sure if I totally understand this plot.

TS stands for Transcriptional Start site and TE stands for Transcriptional End site; these terms have now been defined in the figure legend. This plot shows read signal averaged across all targeted transcripts, demonstrating that the exon probes on the targeted RNAseq panel capture mRNAs from 5' to 3' end.

4) Supp Fig 7 figure legend should indicate that FFT stands for fresh frozen tissue.

Thanks for pointing this out. We have now defined FFT in the figure legend.

Reviewer #2

In the article "Diagnosis of fusion genes using targeted RNA sequencing", the authors describe the application of targeted RNA-Seq to identify fusion genes in cancer samples. Accurate fusion gene detection is a highly relevant topic, both with implications for research and with translational value due to its diagnostic relevance. The authors propose that their targeted RNA-Seq design helps to overcome some of the limitations and has the potential to become a dominant method for detection of gene fusions in a clinical context.

The article is well written and has a clear structure. The authors demonstrate that targeted RNA-Seq is sensitive to identify most gene fusion events. The authors further show that their design can be used to obtain additional insights, for example into the expression levels of fusions genes, non-fusion genes on the panel, and it can profile the immune receptor loci.

The manuscript has a strong focus on the technology, whereas findings from the data are mostly confirmative, or proof of principle. While the authors acknowledge in the discussion that false positives is the major challenge, they do not show a systematic evaluation in the main text. Figure 2 shows examples that suggest the methods is sensitive, but there is no evaluation of specificity, considering false positives for the varying thresholds. The method to identify fusion genes requires manual curation, which prevents a systematic evaluation, and which provides a major barrier for any application outside of a research environment.

Thank you for your encouraging comments about the manuscript. As you indicate, we provide a pre-clinical assessment of targeted RNA sequencing for diagnosing fusion genes (i.e. using retrospective patient samples in a clinical setting). This does not constitute a controlled prospective clinical trial, nor was that our goal.

However, we disagree that manual curation prevents the application targeted RNAseq outside the research environment, as discussed in detail below.

Major comments:**1) Systematic evaluation of sensitivity and specificity**

As the technology is the center of this manuscript, it should be systematically evaluated. Increasing the number of genes compared to FISH will increase both sensitivity and false positives (more genes can be detected). Both needs to be evaluated. What is the false positive rate for single genes (e.g. a direct comparison with single gene methods)? What is the false positive rate when the full array is used? This will be important to understand whether this technology can be applied in a clinical context.

Due to the complexity of the human transcriptome and the absence of patient samples that represent a clear ground truth, it is difficult to provide a simple false positive value for targeted RNAseq as requested by the reviewer. The difficulties in establishing these values are a general feature of NGS-based tests, for which false positive rates can be high and both false negative and true negative rates are typically unknown (Lam HYK *et al.* 2012. Performance comparison of whole-genome sequencing platforms. *Nature Biotech*; Wall JD *et al.* 2014. Estimating genotype error rates from high-coverage next-generation sequencing data. *Genome Research*).

As indicated by the reviewer, this is due, in part, to the breadth of genes tested, each of which will present with different true/false positive/negative rates. Nevertheless, we agree that an assessment of diagnostic performance is required to understand whether targeted RNAseq can be used for clinical diagnosis. Therefore, we provide the following summary of our evaluation of the sensitivity and specificity of targeted RNAseq.

We first evaluated the sensitivity of targeted RNAseq using synthetic spike-in controls that represent fusion genes. These are the only samples for which we have ground truths for true positive, false positive and false negative values. For synthetic fusion genes, we were able to detect fusion genes to a sensitivity of 92% and a false negative rate of 8%, likely driven by the low expression level of the false negative fusion genes. However, whilst synthetic spike-in fusion genes provide a valuable ground truth, they do not encompass the full complexity of the human transcriptome and do not represent every fusion gene targeted, so this evaluation of diagnostic performance is insufficient by itself.

Therefore, we have also evaluated the diagnostic performance of the targeted RNAseq technology using cell lines with known fusion genes. If we are to focus on a single fusion gene, such as *BCR-ABL1*, we find this fusion gene only expressed in cell lines known to harbor the Philadelphia chromosome (K562). Since we do not observe it in any other cell line, this indicates high specificity for individual *BCR-ABL1* fusion genes. This single gene focused assessment is similarly true for any clinically-actionable fusion gene.

However, targeted RNAseq can simultaneously assess a large number of genes. Whilst this is one of its advantages, it also raises the potential for higher false positive rates, as indicated by the reviewer. Therefore, we assessed the performance of targeted RNAseq against a panel of six cell lines that each harbor different fusion genes. In each case, we identified the known fusion genes (i.e. sensitivity = 1 for a clonal sample). We do, however, identify additional fusion genes within these cell lines. Whilst we cannot eliminate the possibility that these represent *bona-fide* low frequency fusion genes present within the cell lines (which often have unstable karyotypes), the low read number for these fusion genes suggests they are false positives.

Finally, we assessed the performance of targeted RNAseq by comparison to patient samples. For these samples, we typically only know the true positive rate of fusion detection (as detected by an alternative test such as FISH), but we do not know their false positive, true negative or false negative rates. Therefore, it is not strictly possible to calculate sensitivity and specificity values. Nevertheless, we do reach 85% agreement between targeted RNAseq and previous clinical fusion detection methods, indicating that with patient samples, targeted RNAseq performs equivalently to technologies that are routinely used in clinical diagnosis.

2) Implementation of a transparent and reproducible informatics workflow

The targeted RNA-Seq technology requires both sequencing and the analysis, and the evaluation should therefore be based on a complete, transparent, and reproducible workflow. In particular, manual curations or filtering should not be included in the evaluation. Manual curation is acceptable to avoid false positives when biological results are reported, but evaluation of a technology should be based on the bioinformatics workflow without manual interference/curation. The workflow should be completely reproducible from the manuscript.

Here, we have provided the complete and reproducible workflow for diagnosing fusion genes. We demonstrate this workflow using two examples: K562 cell line and a CML patient sample (illustrated in Figure 3 below).

Figure 3. Overview of filtering approach to fusion gene identification.

Briefly, we first eliminated fusion gene candidates where neither fusion partner was targeted by the capture panel. While the double capture approach was highly specific, the final sequencing library still contained some off-target sequences. However, we were uninterested in fusions between two non-targeted genes and excluded them from further analysis. Second, we eliminated genes that fell onto a manually curated blacklist containing fusion genes that were consistently

identified across samples (now listed in **Supplementary Table 6** to increase reproducibility of our bioinformatic pipeline). These fusion gene candidates were predominantly fusions between genes with overlapping genomic coordinates, fusions between a parental gene and its pseudogene, or IG/TCR rearrangements. For example, the fusion genes *PPP1CB-SPDYA* and *SPDYA-PPP1CB* were consistently reported, but this is likely due to the overlap in their annotations, as there is no read signal for the majority of *SPDYA* isoforms (see Figure 4 below).

Figure 4. Genome browser screenshot showing one *SPDYA* isoform overlapping the *PPP1CB* gene annotation and K562 read coverage limited to *PPP1CB*. Comprehensive annotations for *PPP1CB* shown in blue, *SPDYA* in green.

After eliminating the blacklisted fusion genes, we then filtered based on read number, as discussed above. We also eliminated fusion gene candidates with fusion partners located on the same chromosome < 10,000 nts apart, as this removed non-blacklisted fusions between genes with overlapping annotations, such as in Figure 4 above. Finally, we shortlisted the fusion genes with the same fusion chromosomal coordinates identified by both *STARfusion* and *FusionCatcher* and subjected them to the manual curation described above, which separated the reported fusion genes (**Supplementary Table 3**) from the false positive fusion genes (**Supplementary Table 7**), based on a whitelist of fusion genes known to be active in each cancer type and the number of supporting reads. As our overall goal was to improve fusion gene diagnosis to expand treatment options and therapeutic outcomes, it is most important that we report informative fusion genes rather than alignment artifacts.

Through the logic-based filtering approach described above, we have developed a data curation pipeline that performs well, demonstrated by the 85% agreement between targeted RNAseq and previous clinical fusion detection. We have provided additional details in the methods section and added **Supplementary Tables 6** and **7** to enable the complete reproducibility of our workflow.

Whilst we agree that omission of manual curation would be ideal, it is not currently practical, and the assertion by the reviewer that evaluation of a technology should be based on a bioinformatics workflow without any manual curation ignores both the high complexity of the human genome/transcriptome and the associated complexity of sequencing-based assays. Furthermore, manual curation is currently required for almost all current routine clinical diagnostic methods. For example, FISH diagnoses requiring manual image analysis performed by a highly trained cytogeneticist.

Manual curation is also required for NGS-based clinical genomics. For example, manual curation is an important aspect of the sequencing analysis pipeline for variant detection and is currently the standard approach. In a recent publication, it was noted that "automated methods for variant refinement are in early stages of development and manual review remains integral to variant identification workflows (Barnell, EK et al. 2018. Standard operating procedure for somatic variant refinement of sequencing data with paired tumor and normal samples. *Genetics in Medicine*). Further, manual review allows individuals to incorporate information not considered by automated variant callers."

Finally, we note that a similar pipeline utilizing manual curation has been implemented by the Molecular Screening and Therapeutics (MoST) clinical trial being performed at the Garvan Institute of Medical Research to great success, demonstrating that manual curation does not prevent clinical implementation.

3) The design could be explained in more detail.

A certain set of genes was selected, but which isoforms/exons/exon-junctions are used? Are known breakpoints included? Are all isoforms captured? If breakpoints are not enriched for, is there a loss of sensitivity to detect fusion genes compared to non-fusion genes?

For each gene, probes against all exons annotated within hg38 were included in the panels, leading to the capture of the whole gene rather than prioritizing specific isoforms. Therefore, known breakpoints were not specifically targeted. Additionally, fusion sequin validation demonstrated that targeted RNAseq successfully captured fusion genes regardless of whether both or only one of the genes was on the panel. We have clarified language in the text to more clearly communicate the design aspect of the panel, as shown below:

"Once the candidate target list was assembled and supplemented with ERCC and fusion sequin sequences, this was sent to Roche for proprietary SeqCap EZ design layout. For the canonical protein-coding genes, biotinylated DNA probes were tiled across all hg38-annotated exons from all isoforms with limited trimming of regions containing repetitive sequences or strong homology to other genes to minimize off-target results."

Specific comments:

Figure 2b, what is the meaning of "On-panel%"?

On-panel % means the percentage of uniquely mapping reads that map to genomic regions targeted by the capture panel. This has been changed to "on-target capture rate" for increased clarity.

Figure 2c, a fair comparison would be fusion reads over non fusion reads for the genes of interest. The total enrichment is much higher due to targeting certain regions, not necessarily due to the enrichment.

We disagree, as we feel that comparing fusion reads in both targeted RNAseq and canonical RNAseq is a fair evaluation of the enhanced performance of targeted RNAseq versus canonical RNAseq. Further, as all exons within each gene are targeted, there is no specific selection or bias towards fusion junctions over non-rearranged splice sites.

Figure 3d: What is the meaning of each point? One patient? What if multiple colors are assigned to one dot?

Each point represents a single patient sample; this has now been clarified in the figure legend. Now that our records have been updated with new information from the tissue bank, we split the point in half for samples where multiple colors were assigned to one dot.

Figure 3e: The numbers are not entirely clear from the legend and table. A systematic analysis of sensitivity and specificity would be good.

Given we do not have a ground-truth understanding of each patient's genotype, we are unable to determine explicit sensitivity and specificity values. Therefore, we are limited to reporting the concordance between targeted RNAseq and alternative clinical diagnoses. This is described in detail above. We have clarified this in the table legend.

Figure 4c: There are 2 fusion gene combinations (as there are 2 promoters). There are also non fusion genes (full length *EZR* and *ROS1* from cells which have intact versions). I don't understand the claim that the fusion gene shows higher expression than the non fusion gene based on these plots, I can only see that fusion gene expression seems to be dominated by the *EZR* expression level. How can I see the expression of non-fusion *EZR* here?

Each box quantifies expression across the canonical gene. In this case, expression of the non-fusion *EZR* is represented by the grey dots. Yes, there are two promoters, so the two alleles that could be expressed are canonical, full length *EZR* and *EZR-ROS1*. At the 5' end of the transcript (blue dots), the quantified expression level will be the sum of the expression of full length *EZR* and the fusion transcript *EZR-ROS1*. The quantified expression at the 3' end of the transcript (grey dots) will only come from full length *EZR* (in theory, if this was a balanced rearrangement, the 3' end of *EZR* could also be expressed as a *ROS1-EZR* fusion transcript, but we detected no evidence of this). Given that the drop in *EZR* read coverage corresponded with the site of the fusion junction, we believe that the grey dots represent expression from full length *EZR*. Since we expect the expression of canonical *EZR* to be the same across all exons, we conclude that the expression difference between the 5' end and the 3' end of *EZR* represents the difference in expression levels between the rearranged *EZR* allele and the canonical *EZR* allele.

Figure 5c/Novel transcriptome features: What is the confidence to detect new exons? Are they supported by multiple reads, multiple samples? How can the authors exclude that these are artifacts? Does the read alignment favor splice site sequences over non-splice site sequences, and could this influence the observed sequence properties for novel exons?

Each new exon is supported by a minimum of 3 junction reads or 20 spanning reads, and the transcriptome assemblies were filtered in such a way to limit spurious novel exon identification. Further, the novel exons have canonical splicing elements flanking the exons and the expected polypyrimidine tracts, suggesting they are bone fide exons. These polypyrimidine tracts are not favored during alignment and therefore do not likely represent alignment artifacts. The novel exons were typically restricted to one or two types of cancer, as would be expected due to tissue-

specific expression.

The number of novel exons discovered here is similar to that previously detected in other targeted RNAseq experiments (Mercer, TR et al. 2011. Targeted RNA sequencing reveals the deep complexity of the human transcriptome. *Nature Biotechnology*; Deveson, IW et al. 2018. Universal Alternative Splicing of Noncoding Exons. *Cell Systems*). It is true that in the absence of secondary validation for every novel new exon, we cannot exclude the possibility that some of these novel exons are artifacts. However, the point of the novel exon discovery performed here was to demonstrate the utility of transcriptome assembly from targeted RNAseq data to discover potential therapeutic peptide targets.

"Diagnosis of fusion genes": To me this sounds misleading as it implies a clinical context. Are the authors referring to "diagnosis using fusion genes" or "detection of fusion genes"? From the abstract it seems like the authors imply both together?

The purpose of this study is to evaluate the use of targeted RNAseq to diagnose fusion genes in patient samples, as an accurate molecular diagnosis can alter disease diagnosis and/or classification. The reviewer is correct in that this study was performed in a clinical context at the Garvan Institute for Medical Research, St. Vincent's Hospital, Sydney using retrospective patient samples.

Reviewers' Comments:

Reviewer #1:

Remarks to the Author:

The authors have nicely addressed all of my concerns except one and that is in their response 5 they state that they have not performed additional intra- and inter-run reproducibility experiments. High reproducibility is a basic pillar of a good test and so I don't understand why these experiments have not been performed. It would be fairly straightforward to do a 3x3x3 experiment, i.e. 3 samples and run them in triplicate within a single run and between three different runs (NOTE--I understand that it might be hard to get all 3 samples in triplicate in a single run for intra-run reproducibility but could run a couple of samples in triplicate in a single run and the remaining sample in triplicate in a subsequent run). I would pick a couple of cases with fusions and one without a fusion. Other than that I think the manuscript is much improved.

Reviewer #2:

Remarks to the Author:

The main points that were raised were to provide a detailed analysis of false positives and to make the bioinformatics steps more reproducible. The authors have provided a detailed response to the reviewer's comments, providing justification and additional details regarding the comments. The authors have in detail explained the limitations to study false positives. Due to these limitations the authors decided not to include a systematic analysis of false positives/specificity in the manuscript. The authors describe the possibility of false positives in the discussion, which helps the reader to be aware of this effect.

The authors have further included several new tables to make the bioinformatics workflow reproducible, which is helpful to understand the different steps, including those that involved manual curation. The authors also note that manual curation is unavoidable.

Most minor points have been addressed. Regarding Figure 4c, the authors write in the response that they do not detect ROS1-EZR fusion genes. The respective section in the manuscript however does not mention this. The expression levels of fusion and non-fusion genes can only be compared if the ratio of cells with the fusion event is known. I recommend that this section in the manuscript includes a statement regarding this effect. Currently, the interpretation that the the 3'end of EZR corresponds to the non-fusion gene is not backed up by the data, it could be that the ROS1-EZR fusion inactivates this gene (the non-fusion gene does not have to be present).

We would like to thank the referees for their comments on our revised manuscript. Below, we addressed each specific comment from the reviews (boxed) and used blue text to indicate changes we made to the manuscript.

Summary of revisions:

- 1) We have performed the 3 x 3 x 3 reproducibility experiment as suggested by Reviewer #1. This resulted in an additional **Supplementary Table 4** and **Supplementary Figure 6** and requisite changes to the text.
- 2) We incorporated reviewer #2's comments into the text regarding **Figure 4c**.

Reviewer #1:

The authors have nicely addressed all of my concerns except one and that is in their response 5 they state that they have not performed additional intra- and inter-run reproducibility experiments. High reproducibility is a basic pillar of a good test and so I don't understand why these experiments have not been performed. It would be fairly straightforward to do a 3x3x3 experiment, i.e. 3 samples and run them in triplicate within a single run and between three different runs (NOTE--I understand that it might be hard to get all 3 samples in triplicate in a single run for intra-run reproducibility but could run a couple of samples in triplicate in a single run and the remaining sample in triplicate in a subsequent run). I would pick a couple of cases with fusions and one without a fusion. Other than that I think the manuscript is much improved.

We have performed the 3 x 3 x 3 experiment suggested above, using 2 patient samples where our initial analysis detected fusion genes (1 CML and 1 AML sample) and 1 patient sample lacking a fusion gene (AML sample). For each sample, sequencing libraries were prepared in triplicate; the 9 samples were captured in triplicate and then sequenced independently on 3 lanes. In each replicate, we detected the expected fusion genes.

This data is now described in the text – paragraph below – and represented in new Supplementary Table 4 and new Supplementary Figure 6.

"To measure the reproducibility of fusion gene diagnosis using targeted RNAseq in patient samples, we selected 3 patient samples – 2 with detected fusion genes, 1 without – and prepared targeted RNAseq libraries in triplicate to assess intra-run variability. These 9 samples were also captured in triplicate and sequenced independently on 3 lanes to assess inter-run variability. We detected the expected fusion genes in all replicates of the 2 positive samples, whilst no fusion genes were detected in any of the negative sample replicates (**Supplementary Table 4**).

We next compared fusion junction read coverage between inter-run and intra-run replicates (**Supplementary Fig. 6a,b**). We observed low variability between inter-run and intra-run replicates with mean coefficient of variations of 0.073 and 0.071, respectively (**Supplementary Table 4**). In addition, we quantified the read coverage for every canonical gene on the capture panel and performed hierarchical clustering to illustrate the high reproducibility in gene expression measurements (**Supplementary Fig. 6c**)."

Reviewer #2

The main points that were raised were to provide a detailed analysis of false positives and to make the bioinformatics steps more reproducible. The authors have provided a detailed response to the reviewer's comments, providing justification and additional details regarding the comments. The authors have in detail explained the limitations to study false positives. Due to these limitations the authors decided not to include a systematic analysis of false positives/specificity in the manuscript. The authors describe the possibility of false positives in the discussion, which helps the reader to be aware of this effect.

The authors have further included several new tables to make the bioinformatics workflow reproducible, which is helpful to understand the different steps, including those that involved manual curation. The authors also note that manual curation is unavoidable.

Thank you for your comments on the revised manuscript. We are pleased that the additional tables improve the reproducibility and comprehension of our bioinformatics approach.

Most minor points have been addressed. Regarding Figure 4c, the authors write in the response that they do not detect ROS1-EZR fusion genes. The respective section in the manuscript however does not mention this. The expression levels of fusion and non-fusion genes can only be compared if the ratio of cells with the fusion event is known. I recommend that this section in the manuscript includes a statement regarding this effect. Currently, the interpretation that the 3'end of EZR corresponds to the non-fusion gene is not backed up by the data, it could be that the ROS1-EZR fusion inactivates this gene (the non-fusion gene does not have to be present).

We disagree that the expression levels of fusion and non-fusion genes can only be compared when the percentage of cells expressing the fusion gene is known; we believe that it is valid to compare the overall expression levels of these transcripts within the sample as a whole. Therefore, when discussing expression levels, we have clarified the text to indicate that expression levels are relative to the sample as a whole and not reflective of intracellular expression levels.

Given that ROS1 is located downstream of EZR on chromosome 6, it is most likely that this fusion gene results from a deletion event and not a balanced rearrangement. Further, a deletion event is supported by our lack of evidence of the ROS1-EZR fusion gene, though we lack the DNA evidence to definitively resolve the genomic structure of this rearrangement. Therefore, we have edited the text to explain that overall expression level of each exon will depend on the sum of expression from multiple transcript types. The relevant section now appears as follows:

"In addition to identifying fusion genes, targeted RNAseq simultaneously measures the expression of all captured genes within each sample¹¹. Initially, we quantified read coverage for each exon and found that abrupt changes in read coverage corresponded to fusion junction locations (Fig. 4c-d). This likely represents the difference in overall expression levels between the fusion gene and the non-fused, canonical alleles, though observed expression levels will depend on the sum of expression of the fusion gene, the inverse fusion gene (in the case of balanced rearrangements), and any non-rearranged alleles."